biophysics

grip force, friction, material properties, cutaneous perception

**Author for correspondence:**
Wouter M. Bergmann Tiest
e-mail: w.m.bergmann.tiest@hr.nl

†Present address: Eindhoven University of Technology, Eindhoven, The Netherlands.

# The influence of visual and haptic material information on early grasping force

Wouter M. Bergmann Tiest[1] and Astrid M. L. Kappers[2,†]

[1]Institute for Communication, Media and Information Technology, Rotterdam University of Applied Sciences, Rotterdam, The Netherlands
[2]Department of Human Movement Sciences, Vrije Universiteit Amsterdam, Amsterdam, The Netherlands

WMBT, 0000-0001-6591-7221

In this paper, we assess the importance of visual and haptic information about materials for scaling the grasping force when picking up an object. We asked 12 participants to pick up and lift objects with six different textures, either blindfolded or with visual information present. We measured the grip force and estimated the load force from the object's weight and vertical acceleration. The coefficient of friction of the materials was measured separately. Already at an early phase in the grasp (before lift-off), the grip force correlated highly with the textures' static coefficient of friction. However, no strong influence on the presence of visual information was found. We conclude that the main mechanism for modulation of grip force in the early phase of grasping is the real-time sensation of the texture's friction.

## 1. Introduction

When picking up objects in a precision grip, people are very good at precisely controlling their grip force to match the friction and load force (weight and inertial force) of the object, with just a small safety margin to prevent slips [1]. This is a highly automated process, where the grip force is adjusted very quickly to the friction between the skin and the object [2]. This process is mediated by tactile afferents in the fingertip skin [3]. The friction arises from a complex interaction between finger and object surface, in which the contact mechanics of the fingertip ridges and the secretion of moisture from sweat glands both play an important role [4]. Regarding the contact mechanics, it was found that changes in shape and size of the part of the contact area that sticks to the surface give information about the onset and direction of slip [5].

Regarding the moisture, it was found that for low contact forces (less than or equal to 3.5 N), the coefficient of friction (CoF) is highest with medium moisture levels, and lower with

low or high moisture levels [6]. For a given texture, the grip force that people use changes when the moisture level (and thus the CoF) changes, suggesting that people accurately perceive the friction and adapt to it [7]. Thus, people react to having dry skin by applying extra grip force [8]. Similarly, grip force is adapted when a texture is modified by applying coatings that change the CoF, confirming that it is friction, not texture, that determines grip force [9]. Incidentally, not only applied grip force but also perceived weight of objects is modulated by changes in the CoF [10]. This is thought to be an effect not of the change in CoF itself, but of the change in grip force, because when people are asked to use a higher grip force than necessary, they also perceive the load as greater [11].

When the object is held stationary or moving at a constant velocity, the load force is identical to the weight of the object. However, when the object is accelerated (e.g. when lifted from the table), inertial forces may add to or subtract from the weight, changing the load force. It has been shown that also in these cases, the grip force is accurately matched to the changes in load force [12,13]. This is not just a reactionary mechanism but also works in an anticipatory way: grip force is modulated in anticipation of changes in load force generated by active movements, regardless of the grip and mode of transport [14].

When picking up an object, the fingers enclose it and start increasing the grip force. The grip force development is programmed based on the expected weight of the object [15]. The increasing grip force causes the fingertips to deform, until the maximum area of contact is reached, at a force of about 1 N [16]. When the hand is forced upward, it exerts a force parallel to the surface on the object, called the frictional force. As the grip force increases further, the frictional force increases with it until the frictional force is large enough to overcome the load force, upon which lift-off can occur. When vision is available, the object and its texture can be recognized in advance, and the correct grip force can be anticipated. Hermsdörfer *et al.* have shown that before object lift-off, grip force was adequately scaled to anticipated weight, based on visual recognition of the object [17]. Also, people have been found to scale their fingertip forces based on visual information about the size and shape of the object, but there is as of yet no evidence that they use visual information about the object's *friction* [18]. This paper tries to find such evidence. Furthermore, when no visual cues are available, the correct grip force must be determined upon contact, either by haptic recognition of the object and its texture, or by direct sensation of the object's weight and friction. In this paper, we investigate how fast people are able to do this. In addition to a fundamental interest in the mechanics of human grasping, information about how people quickly recognize textures by touch is useful for designing haptic feedback systems.

In order to investigate human grasping behaviour in the early phase of the grasp, we asked participants to lift objects with different textures, with and without visual information. The different textures ensure different CoFs, which necessitate different minimum grip forces for the same load force. However, besides friction, another effect of material on grip force has been found: in an experiment in which participants picked up objects of equal mass made of different materials, by a handle, the grip forces were scaled to the *expected* mass of the objects based on a visual estimate of the object size and the material's density [19]. This effect of sensorimotor prediction driven by visual material cues might cause a confound in this experiment: if grip force is not only influenced by the expected friction, but also by the expected mass of the object, it will be difficult to separate these two effects. Fortunately, although the perceptual effect persisted (termed material–weight illusion), the sensorimotor effect disappeared: it was found that after a few lifts, participants started basing their grip force on the *actual* weight of the objects. Thus, we can avoid the confound by disregarding the first lift for each object.

Another possible confound stems from the fact that early grip force was found to be also dependent on the CoF in the *previous* trial. Especially when a surface with a different CoF was presented after a number of trials with the same CoF, participants first used a grip force that was consistent with the previous CoF before correcting to the new CoF [20]. This effect is expected to be smaller when a greater variation in surfaces is presented, but it should be taken into account and its magnitude should be assessed in our measurements.

We measured the grip force during the lift and compared this between the conditions with and without visual information to assess the role of that information. These force measurements have been reported before in conference proceedings [21]. In addition, in this paper, the inertial forces were estimated from movement tracker measurements. This allowed the ratio of grip force and load force to be calculated, which is a measure for the perceived slipperiness of the texture: for the same load force, a higher grip force is necessary to prevent slipping when the texture is more slippery, and vice versa. The calculated ratios for all textures were compared to their static friction coefficients, which were

measured separately. These friction measurements are reported first, followed by the experiment with human participants.

# 2. Friction measurements

## 2.1. Material and methods

Six steel blocks (40 × 40 × 40 mm) were covered with different materials: leather chamois, ribbed cloth, coarse sandpaper, metallic-like plastic film, transparent acrylic glass and wood-like plastic film. The metallic-like and wood-like textures were plastic self-adhesive films mimicking the structure of metal and finished wood, respectively. The stimuli are shown in figure 1. The textures were visually and haptically quite distinct and easy to identify. It was clear that the blocks were only covered by the materials and did not consist of them all the way through. Thus, it was not suggested that the blocks had different densities.

To measure the coefficient of static friction, we used a device described in Platkiewicz *et al.* [22]. In this device, the surfaces were placed horizontally on a tribometer consisting of three force transducers: two measuring the normal force (both Kistler 9313AA1) and one measuring the tangential force (Kistler 9217A). The transducer signals were amplified and digitized using a 16-bit ADC at 10 kHz. Before the start of each measurement, the charge amplifiers were nulled to minimize the effect of drift. After the measurement had started, the experimenter placed a finger on the block with a normal force of about 5 N. This force level was achieved following a few practice runs. Then, the experimenter increased the tangential force until the finger started sliding over the surface. This method is similar to the one used by Platkiewicz *et al.* [22]. For each block, two measurements of 5 s each were taken.

It should be noted that the CoF for a finger touching a specific material can differ greatly from person to person and also depends strongly on humidity and other circumstances [4–6]. From that perspective, it might be preferable to use separate measurements for each participant, instead of a single set measured with the experimenter's finger. However, that would make it impossible to pool data per material, greatly hampering the statistical analysis. Therefore, the choice has been made to use a single set of measured CoFs.

## 2.2. Results

The ratio $\mu$ of tangential force over normal force was plotted as a function of time for each measurement. An example is shown in figure 2. The maximum value of $\mu$, just before the finger started sliding, was taken as the coefficient of static friction. The measured values are reported in table 1. In subsequent analyses, the average of the two measurements for each sample was used.

# 3. Grip force measurements

## 3.1. Material and methods

### 3.1.1. Participants

Twelve right-handed people (four men, eight women; aged 20–27 years) took part in the experiment. They reported no tactual or uncorrected visual deficits. After receiving instructions, but before the experiment started, they provided written informed consent. They received a payment of €8 for their time (1 h). The experiment was approved by the Ethics Committee of the Faculty of Human Movement Sciences, Vrije Universiteit Amsterdam.

### 3.1.2. Stimuli

The same six steel blocks from the friction measurements were used. The blocks were quite heavy (0.500 kg), in order to encourage participants to not use more grip force than was necessary to prevent slipping. With lighter blocks, participants might have been tempted to use the same, larger-than-necessary grip force for all materials, ignoring material differences. As it was clear that the blocks were only covered by the different materials and consisted internally of the same material,

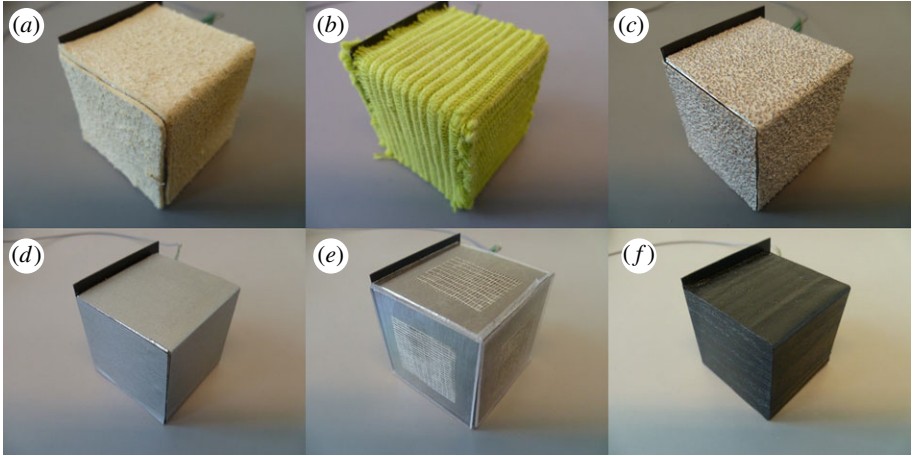

**Figure 1.** The six blocks clad with different materials. (*a*) Leather chamois, (*b*) ribbed cloth, (*c*) coarse sandpaper, (*d*) metallic-like plastic film, (*e*) transparent acrylic glass and (*f*) wood-like plastic film. The force sensors can be seen sticking out on the top left side.

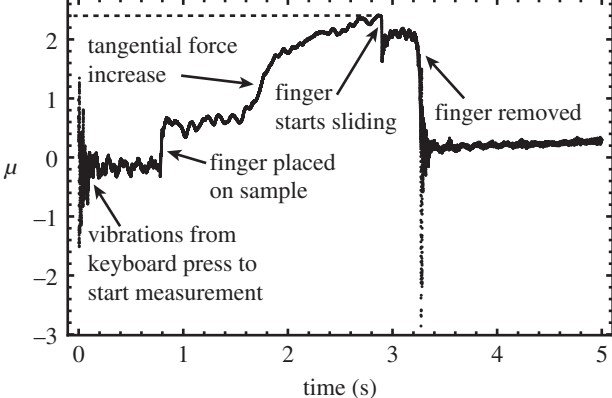

**Figure 2.** Example of a friction measurement with the tribometer for the wood-like texture. The ratio $\mu$ of tangential force over normal force is plotted as a function of time. The different phases of the measurement are indicated. The dashed line indicates the maximum value of $\mu$, in this case, 2.4.

there was no suggestion that the blocks had different masses. For this reason, we expect no large effects of the material–weight illusion [19].

On the thumb side, between the steel block and the covering material, a $38 \times 38$ mm force sensor (FSR 406, Interlink Electronics, accuracy 0.01 N, repeatability $\pm 2\%$ [23]) was inserted to measure grip force. Using this sensor, the grip force perpendicular to the surface could be measured. Rotational forces could not be measured, but since participants were instructed to perform a vertical lift, these will likely be insignificant, especially in the early phase of the lift.

To detect the moment of lift-off, another force sensor was placed underneath a 15 mm high platform, on which a stimulus could be placed before being grasped by the participant. This last sensor was not calibrated, as it was only used as a switch to detect the presence of a stimulus on the platform. Each sensor's resistance was measured separately by means of a voltage divider with a 10 kΩ resistor and a 5 V bias voltage and digitized at 1 kHz using a 12-bit ADC board (PCI-1200, National Instruments).

On the back of each stimulus, an infrared LED marker was placed, which enabled tracking the position and thus determining the acceleration of the stimuli. Similar markers were placed on the thumb and index fingernails of the participant. The markers were tracked at 100 Hz using an NDI Optotrak Certus 3D motion tracking system, with an accuracy of approximately 0.2 mm.

### 3.1.3. Calibration

Because the materials covering the force sensors might deform in different ways when compressed, it was necessary to calibrate the sensor for each stimulus separately. Also, the force sensors used in this

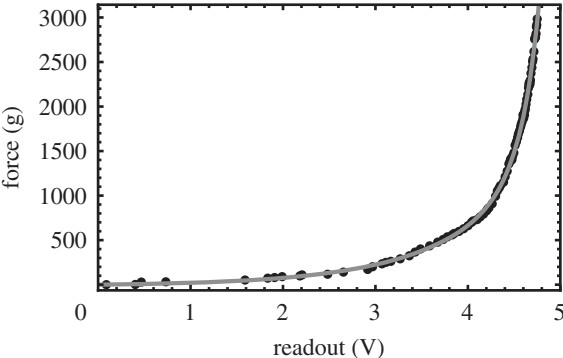

**Figure 3.** Example of a calibration measurement of the force sensor for the wood-like texture. The dots are the measured data points, and the grey line is a fit to the data.

**Table 1.** Measured coefficients of static friction.

| texture | Meas. 1 | Meas. 2 | average |
|---|---|---|---|
| chamois | 2.3 | 2.7 | 2.5 |
| cloth | 3.4 | 3.4 | 3.4 |
| sandpaper | 4.6 | 4.3 | 4.4 |
| metallic | 2.2 | 2.3 | 2.3 |
| acrylic | 3.1 | 3.6 | 3.3 |
| wood-like | 2.4 | 2.7 | 2.6 |

experiment are actually pressure sensors, and their response is therefore not independent of the size of the contact area. Since different participants have differently sized thumbs, the sensors' response might also be participant-dependent. Thus, the calibration was performed for each stimulus and each participant separately, as follows: the stimulus, oriented with the sensor on top, was placed on a digital weight scale (Mettler Toledo Spider A6, precision 0.001 kg), which was read out by a computer at a rate of about 15 Hz. The participant was asked to place his/her thumb on top of the stimulus, while the measured force was displayed in a graph on a computer screen. S/he then had to gradually increase the pressing force until the indicator reached the right-hand side of the graph, corresponding to 3 kg (29 N). This was repeated for each of the six stimuli. To the measured force $F$ (in g) as a function of the measured voltage over the sensor $V$ (in V), the empirical function

$$F(V) = a(\exp(bV) - 1) + c(\exp(dV^2) - 1)$$

was fitted, with $a$, $b$, $c$ and $d$ free parameters. An example is shown in figure 3. A total of 72 of such fits were performed (12 participants × 6 stimuli), and the coefficient of determination $R^2$ was greater than 0.994 for all. In further analyses, the force was expressed in N by multiplying by 0.0098. To assess the variance within the calibrations, we looked at the spread of forces corresponding to a readout voltage of 4.5 V, which is close to full scale. On average, this corresponded to a force of about 12 N. The coefficient of variation (standard deviation divided by mean) was $9 \pm 3\%$ between participants (averaged over stimuli) and $14 \pm 3\%$ between stimuli (averaged over participants). This shows that it was worthwhile doing the calibrations separately for each stimulus and participant.

### 3.1.4. Procedure

The experiment consisted of two conditions: one blindfolded and one sighted. The blindfolded condition was always performed first, so that participants could not form a visual–haptic association before starting the blindfolded condition. During the visual condition, the stimuli not used in a particular trial were hidden from view by a curtain. Before the experiment proper began, the blindfolded participant was allowed one or two practice trials using the same stimuli as used in the experiment. Having been exposed to some of the stimuli before the experiment probably reduced the sensorimotor effect associated with the material–weight illusion during the experiment.

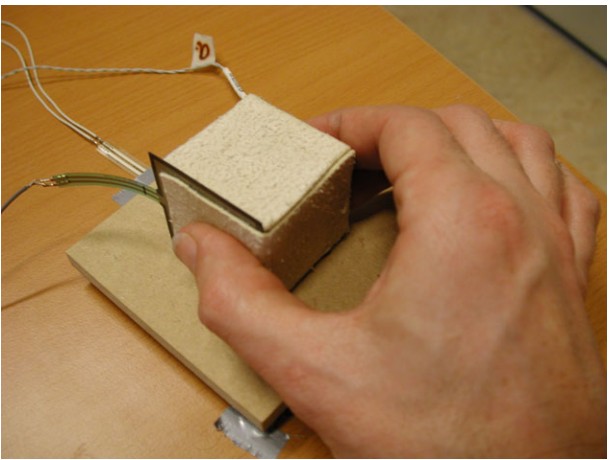

**Figure 4.** Photograph of a stimulus being grasped in a precision grip.

Each condition consisted of 60 trials, 10 for each stimulus, in random order. Each trial, the experimenter placed the stimulus on the platform. Then, an auditory signal was given, prompting the participant to close the fingers around the stimulus and lift it a few centimetres, and then set it down again. The participant had to use a precision grip with the thumb on one side, and one or more fingers on the other side of the stimulus, as shown in figure 4. The elbow rested on the table during the whole lift, and rotational movement was kept to a minimum. Since only the thumb force was measured, no restrictions were put on the finger placement on the other side. No instructions were given about the amount of force to be used. For each trial, the grip force was recorded for 2 s (2000 samples) and the position of the stimulus and the participant's fingers for 3 s (300 samples). On average, the two conditions together were completed in $40 \pm 4$ min (mean $\pm$ s.d.).

### 3.1.5. Analysis

The first trial for each material was discarded in both conditions for all subjects, in order to avoid the sensorimotor effect associated with the material–weight illusion. Using the individual calibrations, the measured voltage signals from the force sensors were converted to grip forces. For each participant, the maximum grip forces averaged per material were correlated with the reciprocal measured friction coefficients. The correlation coefficients $R$ were transformed to $z$-scores using Fisher's transform and subsequently averaged. The average $z$-score was transformed back to an $R$-value.

From the measured vertical positions of the stimuli, the vertical acceleration was calculated using a second-order central discrete differentiation method, and this was multiplied by the mass of the stimulus and added to its weight to obtain an estimate of the load force:

$$F_{\text{load},i} = m \frac{z_{i-1} - 2z_i + z_{i+1}}{\Delta t^2} + mg,$$

where $z$ is the vertical position, $\Delta t$ the sample interval (0.01 s), $m$ the stimulus mass (0.500 kg) and $g$ the gravitational acceleration (9.82 m s$^{-2}$). Note that this yields a load force equal to the weight of the stimulus whenever it is stationary, also when it is resting on the table, when the actual load force should be zero. Therefore, this equation only holds when the stimulus is in the air. For this reason, we are unable to measure the load force prior to lift-off with this method. However, for our main analysis, we are interested in the early grip force, which is available also before lift-off.

For each trial, the two force profiles were synchronized by identifying the moment of lift-off in the grip force and load force profiles separately. In the grip force profile derived from the force sensors, this moment corresponded to the signal from the sensor underneath the platform crossing a certain threshold. In the load force profile calculated from the position signal, this moment corresponded to the vertical position first exceeding a value of 0.2 mm above the baseline position, which was derived from the average of the first 10 samples (0.1 s) in that trial. An example of the synchronized force profiles is shown in figure 5. Note that the grip force starts being applied some 0.2 s before lift-off. The onset of the grip force was determined as the moment the grip force first exceeded 0.01 N.

The grip force was divided by the load force to yield the grip force/load force ratio. From the force profiles, the following quantities were derived: maximum grip force, grip force 10 ms after onset,

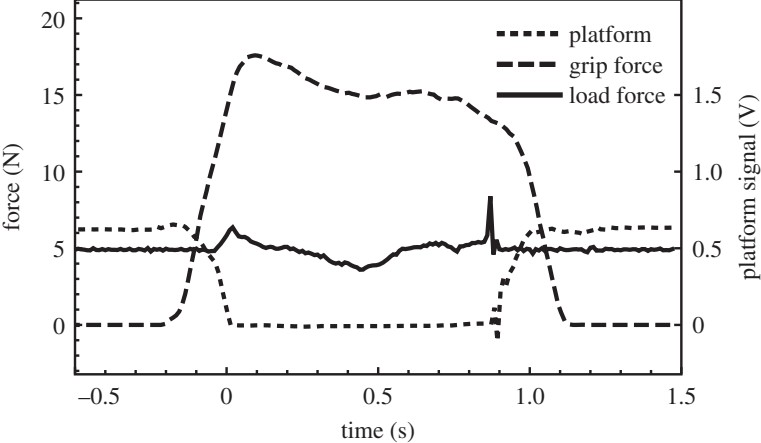

**Figure 5.** Example of the two force profiles for a stimulus being picked up and set down again in an individual trial. 1440 of such trials were recorded. Time zero corresponds to the moment of lift-off, as derived from the 'platform signal' (dotted). The grip force (dashed) is measured under the thumb, while the load force (solid) is derived from the vertical acceleration of the stimulus and its weight. For this reason, the load force is shown as 4.9 N when the stimulus is resting on the platform.

maximum load force and maximum grip force/load force ratio. The time point of 10 ms was chosen because it is too soon after contact was first made for any haptic information to be involved, but still after contact is established, enabling grip force measurement. Thus, it is the earliest point in time at which an influence of visual information might be expected, but not of haptic information, allowing a distinction to be made between the two.

The effect of material and the presence of visual information on these quantities were determined using repeated-measures ANOVAs. When sphericity was violated according to Mauchly's test, Greenhouse–Geisser correction was used. Furthermore, these quantities, averaged over participants, were correlated with the materials' coeffcients of static friction from the friction measurements. The reciprocal of the CoF was used, because then a linear relationship is expected. Lastly, the significance of the Pearson correlation coefficient $R$ between grip force and reciprocal CoF was assessed for every time step in the pre-lift-off phase for each participant. These $R$ values were transformed to $z$-values using the Fisher transformation and averaged over participants. Since the transformed values are normally distributed with a standard deviation $\sigma = 1/\sqrt{n-3}$, the critical $z$-value corresponding to a value of $p = 0.05$ for $n = 6$ data points is given by

$$z_{\mathrm{crit}} = \sqrt{\frac{2}{3}}\, \mathrm{erfc}^{-1}(0.05) \approx 1.13,$$

where $\mathrm{erfc}^{-1}$ is the inverse complementary error function, i.e. the inverse of $(1 - \mathrm{erf})$. Using this value, we can see when the correlation between grip force and CoF becomes significant. It should be noted that this is not a discrete transition, but it provides an indication of how quickly the grip force is adapted to the materials' friction.

The analysis in this study focused mainly on the maximum grip force and its correlation with the CoF. Other characteristics, such as grip force rate, load force rate or time-to-peak are not considered. Analysis of these characteristics might yield other insights, but does not contribute to answering our main research question.

## 3.2. Results

Before going into the results for the different materials, it is worthwhile to look into the effect of the material used in the previous trial. This was done by selecting those trials that were preceded by a more slippery material (increasing CoF) and contrasting them with those trials that were preceded by a less slippery material (decreasing CoF). Pooling all materials together, the maximum grip force was found to be significantly *smaller* with an increasing CoF compared to a decreasing CoF, both in the blindfolded ($t_{11} = -6.0$, $p = 0.000079$) and sighted conditions ($t_{11} = -4.9$, $p = 0.00051$), as seen in figure 6. A similar effect was found for the maximum grip force rate: $t_{11} = -4.6$, $p = 0.00075$ (blindfolded) and $t_{11} = -3.4$, $p = 0.0061$ (sighted), but not for the grip force at 10 ms ($t_{11} = -0.74$, $p = 0.47$).

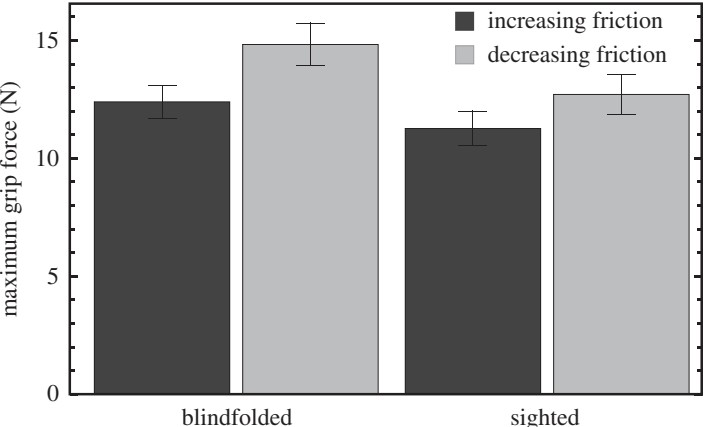

**Figure 6.** Maximum grip force averaged over trials and participants, with all materials pooled together, split between trials preceded by a more slippery material (increasing friction) and trials preceded by a less slippery material (decreasing friction). The error bars indicate the standard error of the sample mean.

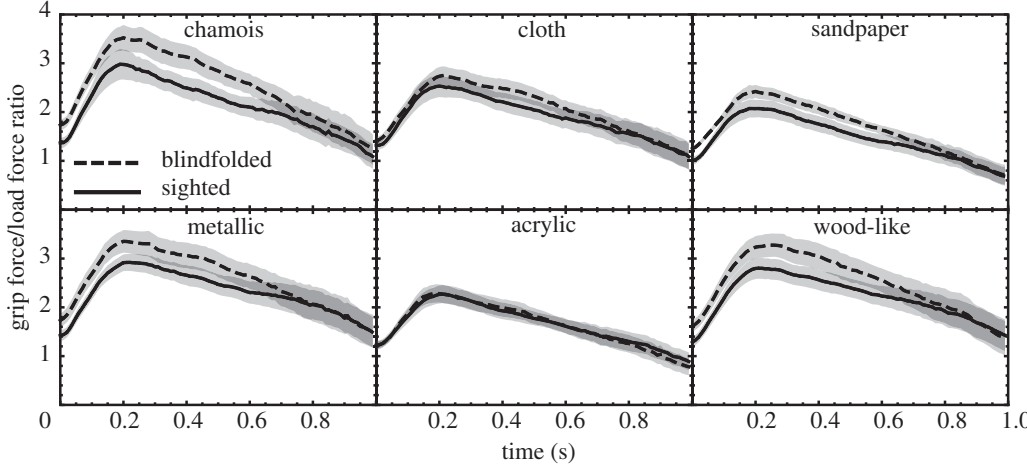

**Figure 7.** Grip force/load force ratios averaged over trials and participants for the first second of the lift. The time axis is referenced to the moment of lift-off. The profiles do not start at zero because the grip force has already started being applied in the phase before lift-off. The shaded area around each curve indicates the standard error of the sample mean.

The *grip force/load force ratios* for the different materials and the two conditions, averaged over trials and participants, are shown in figure 7. A $6 \times 2$ repeated-measures ANOVA on the grip force/load force ratio, averaged over the time interval between lift-off and reaching the maximum, showed a significant effect of material ($F_{1.9,21} = 15$, $p = 0.00011$) and of condition ($F_{1,11} = 6.0$, $p = 0.032$). Also, there was a significant interaction effect ($F_{5,55} = 3.0$, $p = 0.019$). This indicates that the average ratio differed between the two conditions for some, but not all, materials. Paired *t*-tests showed that this was the case for sandpaper ($t_{11} = 3.7$, $p = 0.0036$), metallic-like plastic film ($t_{11} = 2.9$, $p = 0.015$) and wood-like plastic film ($t_{11} = 3.0$, $p = 0.013$), but not for the others ($t_{11} < 2.2$, $p > 0.052$).

Since all blocks had the same mass, the *maximum load forces* were very similar for the different materials, but there still was a significant effect of material ($F_{3.0,32} = 7.4$, $p = 0.00069$). On average, the maximum load force was 6.4 N for the blindfolded condition and 6.7 N for the sighted condition. This difference was just significant ($F_{1,11} = 5.1$, $p = 0.045$), showing that in the sighted condition, the stimuli were lifted with a somewhat greater acceleration.

The *maximum grip force* for the different materials and the two conditions, averaged over trials and participants, is shown in figure 8. As expected, the maximum grip force was highly correlated with the reciprocal of the coefficient of static friction: $R_4 = 0.91$, $p = 0.0092$ for the blindfolded condition, and $R_4 = 0.97$, $p = 0.00037$ for the sighted condition. This last correlation is illustrated in figure 9.

This high correlation shows that grip force during the lift was accurately modulated by the material's friction, confirming earlier work and indicating that the stimulus set was indeed suitable

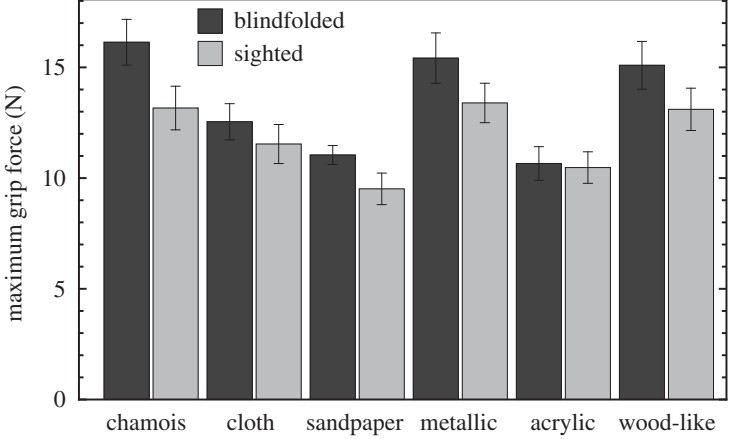

**Figure 8.** Maximum grip force averaged over trials and participants. The error bars indicate the standard error of the sample mean.

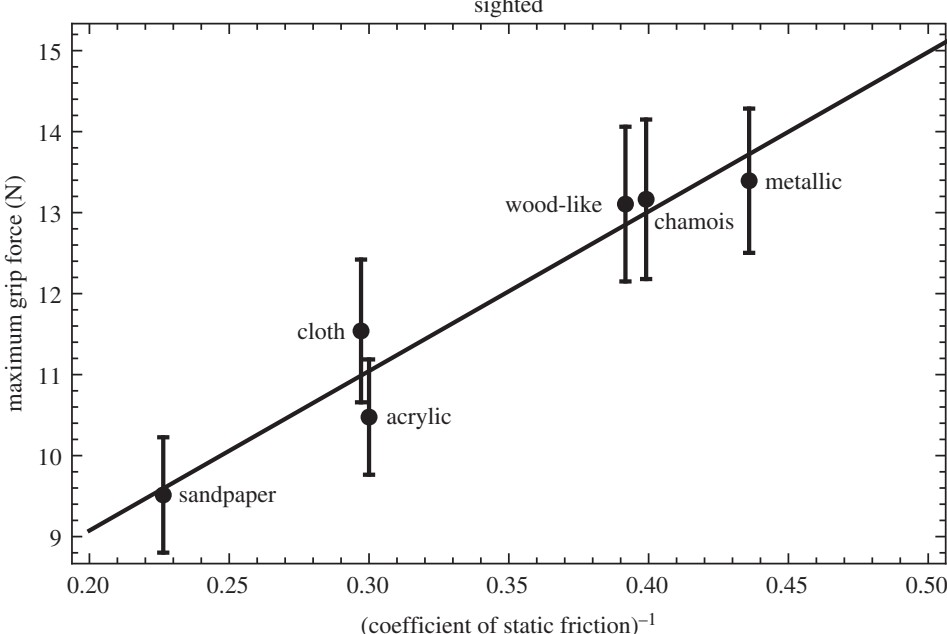

**Figure 9.** Average maximum grip force in the sighted condition plotted as a function of the reciprocal coefficient of static friction for the six materials. The error bars indicate the standard error of the sample mean. The solid line is a linear fit to the data with the equation $F_{\mathrm{grip,max}} = 19.7/\mu + 5.14$.

for studying grip force differences. It also reinforced our decision to use a single set of friction values for all participants, enabling us to pool their data, thereby strengthening the statistical analysis. However, our main interest was in the early grip force, before lift-off. Therefore, we looked at the *grip force at 10 ms* after initial contact was made. We hypothesized that in the sighted condition, with visual information available, the stimulus could be recognized beforehand and the grip force could be adjusted to the remembered friction right from the onset. Thus, we expected an effect of material on early grip force in the sighted condition. In the blindfolded condition, no information is available beforehand about the friction of the stimulus, so we did not expect an effect of material on early grip force in that condition. Surprisingly, the reverse was found: a significant effect of material on grip force at 10 ms after onset in the blindfolded condition ($F_{1.7,19} = 7.5$, $p = 0.0052$), and no effect in the sighted condition ($F_{2.3,25} = 2.0$, $p = 0.17$; separate one-way repeated-measures ANOVAs in the two conditions). Although there were differences between the grip forces at 10 ms for the different materials in the blindfolded condition, they were, however, not correlated with the coefficients of static friction ($R_4 = -0.032$, $p = 0.96$).

In order to determine at what moment in the early phase of the grasp the grip force becomes correlated with friction, the correlation coefficient was calculated for every 1 ms time step and

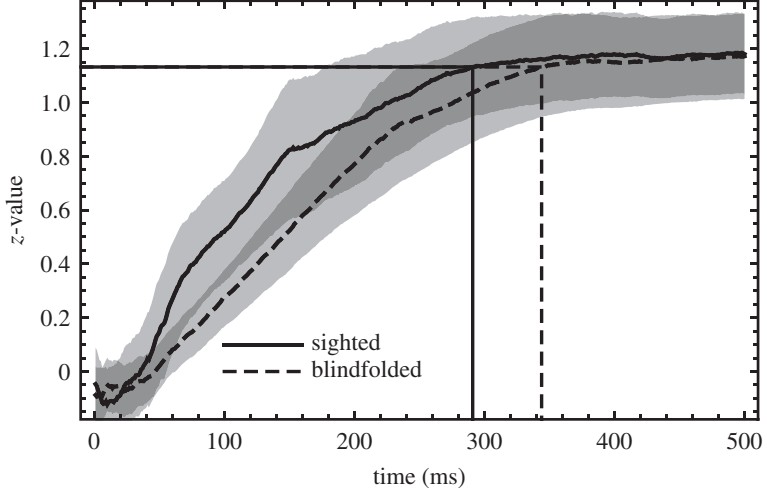

**Figure 10.** z-values transformed from Pearson correlation coefficients between the average grip force and the reciprocal coefficient of static friction, plotted as a function of time since contact onset. The critical level, corresponding to $p = 0.05$, is indicated by the horizontal line. The curves cross this level at the moment the z-value, and thus the underlying correlation, becomes significant. The shaded areas correspond to the standard error of the sample mean.

transformed to a z-value, as shown in figure 10. The curves cross the critical level of $z = 1.13$, $p = 0.05$ at 344 ms and 291 ms after contact onset in the blindfolded and sighted condition, respectively.

## 4. Discussion

As can be seen in the example of figure 5, the grip force is dynamically adapted to changes in the load force that are the result of the stimulus being accelerated and decelerated vertically. However, the ratio between the two is not constant over time during the lift, as can be seen in figure 7. Even though the ratio is already high enough to prevent slipping (lift-off has occurred, after all), it is still increasing for about 0.2 s after lift-off. This might be ascribed to a kind of inertia effect: the participant does not stop applying more grip force immediately after the maximum in load force (vertical acceleration) has been reached, but continues for some time. When the maximum ratio is reached, the participants use an ample safety margin: based on the measured friction, a ratio ranging between 0.23 and 0.44 is necessary to prevent slipping, yet more than five times as much force is applied. Already right after lift-off, this safety margin factor is about 1.6 for the sighted condition and about 2.7 for the blindfolded condition. This is quite a bit higher than was reported in earlier work [1], perhaps due to the fact that in the present experiment, no instructions were given regarding the force to be used.

When the grip force/load force ratio is averaged over the time interval between lift-off and reaching the maximum, there is an effect of condition: for sandpaper, and for metallic-like and wood-like plastic film, the ratio is significantly larger in the blindfolded than the sighted condition. Apparently, people employ a larger safety margin for those materials when blindfolded, possibly because their lack of vision made them feel a little more unsure.

Previous work has found a clear effect of the CoF in the previous trial on early grip forces [20]: when the same stimulus is presented repeatedly, followed by a different stimulus, the participant expects this stimulus to be the same as the previous ones and keeps using the same grip force. In the present experiment, however, stimuli are different almost every time, so no such expectation effect occurs. Rather, a reverse effect was found: when a more slippery material was preceded by a less slippery one, a smaller grip force was used compared to the other way around (figure 6). This is consistent with the expectation that occurs when materials are different almost every time. In this case, simply due to the statistics of the stimulus set, a slippery material is more likely to be preceded by a less slippery one than by a more slippery one. Therefore, the expectation of the participant after having felt a less slippery stimulus is that of a more slippery stimulus, resulting in a lower grip force. So, interestingly, although the effect seems to be in the opposite direction compared to the literature [20], it is evidence for the same explanation based on the participant's expectation.

The fact that there was a small but significant effect of material on maximum load force shows that some materials were systematically lifted more quickly than others (greater vertical acceleration). Even

though it was clear to the participants that all stimuli had the same mass, it might have been that people subconsciously expected some blocks to be heavier than others based on the texture, and applied more lifting force to those. This might have been caused by remnants of the sensorimotor effect associated with the material–weight illusion, despite the first trial having been discarded for each material [19]. This is in line with more recent findings by the same group, showing an effect of expected mass on maximum load force that remains significant for four lifts, compared to just one or two for the effect on maximum grip force [24]. However, the size of the effect in the current study is quite small (no more than approx. 5 % load force difference between the extremes, compared to over 100 % difference in grip force at 10 ms, and 50 % difference in maximum grip force), so the material–weight illusion is not expected to have an influence on the early grip force results. It was also found that the blocks were lifted somewhat more quickly in the sighted condition than in the blindfolded condition, possibly due to participants moving a little more hesitantly when blindfolded or because they had had more practice by the time they reached the sighted condition.

In this study, CoFs were not measured for each participant separately, but only with the experimenter's finger, so as to be able to pool the data per material. Another way to do this would have been to measure the friction per participant separately, but average the data over participants. This was not done, as averaging across participants would cause a decrease in signal-to-noise ratio. As shown in table 1, only two CoF measurements were performed for each material. Although this is a limitation, this was deemed acceptable as the difference between the two measurements for each material is considerably smaller than the range of measured CoFs.

The high correlation between maximum grip force and reciprocal CoF shows that also for our stimuli, participants accurately match their grip force to the material's friction, confirming earlier findings [1,2,9]. This means that our stimulus set contains enough variation in friction to elicit variation in grip force, making it suitable for characterizing the influence of material and visual information on early grip force. We hypothesized that anticipatory force scaling might show up very early in the grip force profile when visual information about the stimulus identity was available beforehand, but not when this information was absent. Surprisingly, there was a significant effect of material on grip force at 10 ms after contact was made in the blindfolded condition, whereas there was not in the sighted condition. This time scale is too short for a cognitive process to be involved, such as haptic recognition of the stimulus texture and retrieving its friction from memory. It is also too short for an automatic process in which grip force is immediately adjusted to perceived friction, based on the detection of microslip. The correlation analysis revealed that this effect was not related to the materials' CoFs. It is not clear what could have caused this effect at such an early time in the force profile. We speculate that the mechanical properties of the materials situated between thumb and force sensor might be responsible. Perhaps more rigid materials transfer the applied force more directly to the force sensor, causing a faster response. Also, the way the finger impacts on the surface may cause differences in measured force. Indeed, somewhat thicker, more compliant materials (chamois and cloth) show relatively low grip forces at 10 ms, while thinner materials (sandpaper and metallic-like plastic film) show relatively high forces. On the other hand, acrylic glass, the most rigid material, shows a relatively low force. Thus, this explanation cannot account for all facts. Moreover, it also does not explain the absence of a similar effect in the sighted condition.

The correlation analysis shown in figure 10 does show that there is a relation between friction and grip force, starting at approximately 50 ms after onset and becoming progressively stronger until significance is reached at 344 ms and 291 ms for the blindfolded and sighted conditions, respectively. These numbers should not be interpreted as exact, as they are based on the quite arbitrarily chosen value of $p = 0.05$, but they do provide an indication of how early in the grasp the grip force is already modulated by the texture. The correlation for the blindfolded condition seems to lag somewhat behind that for the sighted condition, suggesting the possibility of visual information helping to speed up the modulation. However, the difference is minimal, as evidenced by the large overlap of the ranges of standard error at this point. Since there is no significant correlation between grip force and CoF in the first 100 ms after contact is made, we have no clear indication of the use of visual information, available long before contact, to retrieve a friction estimate from memory, and therefore we conclude that the main cause for the anticipatory modulation of grip force is haptic sensation of the texture and that the role of visual information is small.

A similar result was found in a recent study in another area of sensorimotor control, the planning and execution of motion. In an experiment where participants had to reach towards a target that could either stay still or move sideways during the motion, no difference in correction latency was found between a condition in which only haptic information was present during the move versus a condition in which

both visual and haptic information were present [25]. Similar to this study, this shows that visual information does not play a large role when haptic information is available.

One alternative explanation for the lack of evidence for the use of visual texture information to modulate grip force might be due to the experimental design. All participants first completed the blindfolded condition before starting the sighted condition, because we did not want them to prematurely form a visual–haptic relationship regarding the textures. It might be that after getting used to being deprived of visual information in the blindfolded condition, they continued disregarding visual information in the sighted condition, whereas in a real-life situation, they would use visual information. However, based on the fact that they have been using visual information their whole life, it seems unlikely that a few minutes of wearing a blindfold could make them relinquish it, but we cannot rule it out.

## 5. Conclusion

We conclude that grip force development during the loading phase of grasping an object is not only modulated by the anticipated weight of the object, but also by the real-time sensation of the texture's friction. The fact that this takes place within a few hundred milliseconds after contact is made, illustrates the speed of the haptic perceptual system.

Ethics. The experiment was approved by the Ethics Committee of the Faculty of Human Movement Sciences, Vrije Universiteit Amsterdam (Programme Haptic Perception, 2012-55). After receiving instructions, but before the experiment started, the participants provided written informed consent.
Data accessibility. The dataset supporting this article is available at http://hdl.handle.net/10411/C5QQGF.
Authors' contributions. W.M.B.T. and A.M.K.L. conceived the study, performed the data analysis and wrote the article. W.M.B.T. built the apparatus and performed the experiments.
Competing interests. We declare we have no competing interests.
Funding. This work has been partially supported by the European Commission with the Collaborative Project no. 248587, 'THE Hand Embodied', within the programme 'Cognitive Systems and Robotics'.
Acknowledgements. We would like to thank Séréna Bochereau (Institut des Systèmes Intelligents et de Robotique, Université Pierre et Marie Curie, Paris, France) for her help with the friction measurements.

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
