## [Reviewer comments · Royal Society Open Science]

Review History

RSOS-172175.R0 (Original submission)

Review form: Reviewer 1

Is the manuscript scientifically sound in its present form?

No

Are the interpretations and conclusions justified by the results?

No

Is the language acceptable?

Yes

Is it clear how to access all supporting data?

Yes

Do you have any ethical concerns with this paper?

No

Have you any concerns about statistical analyses in this paper?

No

Recommendation?

Major revision is needed (please make suggestions in comments)

Comments to the Author(s)

This short paper reports two experiments – one addressing the static friction of a set of different objects, when shear force is applied to the point of slip by a human thumb – and the second comparing grip forces when participants grip and lift these objects with and without vision. The main result of the paper is to report differences in the impact forces (which I think is a better description of what the authors measure) and of grip/load force ratios after lift-off.

The paper follows in the line of previous work on this topic, but I think the question asked here (the effect of visual vs non-visual lifting, without in the latter case prior knowledge of the object), is novel. However, there are issues with the design that limit my enthusiasm.

First, measures of static friction are made by one of the experimenters. The authors report how this measure is influenced by e.g. skin moisture etc and therefore also by individual characteristics of the participants' fingers. Thus there may differences in individuals' friction coefficients that might be important. It is a shame they did not measure this for each participant. The authors claim that the rank order of the friction estimate is unlikely to differ. This is somewhat undermined by their own data, showing for example that the variation in friction estimates (across only 2 measures) is as large as the difference in average friction between surfaces. Thus the rank order could well differ – in fact it does differ even between two measurements by the same person.

Second, they have no data on load force prior to the object lift-off. This is a critical phase of the grasping action. By and large, once lift off has occurred, the grip/load force ration is sufficiently large that there is minimal difference across surface materials. However it seems quite likely that the participants, especially when deprived of vision, might modulate grip prior to lift-off, by sensing slip. This would be unmeasured in their data.

Third, although a minor point, the load forces are estimated by integration of positional data sampled at 100 Hz into acceleration estimates, and this is likely to be noisy. It may also be important that there is no measure of object rotation, and one would like to be able to rule out any rotational slips, again, especially in the non-vision conditions. It would be interesting also to report more directly on the load force, or object motion (eg acceleration), rather than only showing grip/load ratios or statistics of load force differences.

Finally, a lot of the paper is devoted to measures of “grip” force 10 ms after contact. I am sure that the force will be changing rapidly at this early moment, as it will be affected by the impact force (depending on speed of finger contact) and by object compliance, as the authors do acknowledge. While I accept that there is a danger of confounding later measurements by feedback responses, I think that there could easily be changes in “grip” for 10's of ms, driven only by these impact effects.

Minor comments

Please justify the use of the reciprocal of CoF rather than minus-CoF (if a positive correlation is necessary – why? – it would sufficient to invert the sign).

Please explain what complementary error function is (equation on page 6).

Review form: Reviewer 2 (Nicholas Holmes)

Is the manuscript scientifically sound in its present form?

Yes

Are the interpretations and conclusions justified by the results?

Yes

Is the language acceptable?

Yes

Is it clear how to access all supporting data?

Yes

Do you have any ethical concerns with this paper?

No

Have you any concerns about statistical analyses in this paper?

Yes

Recommendation?

Accept with minor revision (please list in comments)

Comments to the Author(s)

The authors have presented an excellently-written manuscript. I could not find a single typo! More importantly, the background, methods, results, and conclusions are all very clear and easy to read. Probably the best-written manuscript I've reviewed in several years. I am not an expert in this field, but I learnt a lot from this very-clearly written manuscript.

I only one minor comment, which refers to the parametric analysis of correlation coefficients. I'm sure the proposed changes will improve the manuscript, and make the between-condition differences stronger, but some of the conclusions about the timing may change a bit:

Minor comments

Page 7, line 42 - R-values need degrees of freedom - 4?

It would be better, in my opinion, here to calculate an r-value for each participant, to convert all the participants' r-values to Z-values (Fishers r-to-Z), then run a t-test on the Z scores, testing against zero. It is possible (though unlikely) that a correlation across (the means across participants for each of) the materials could arise, but that each participant does not show this correlation individually, and the group mean correlation is not significantly greater than zero. Better to test the within-participant correlation and express with confidence interval/standard error (on the Z-scores). Mean Z-scores could then be converted back into a group r-value if needed.

Figure 10 & statistics on page 8 - r-values need to be converted to Z values before any analysis: r is non-linear (and non-parametric, with bounds at -1 and 1), so the standard error bars are not representative and suffer from ceiling and floor effects, and the p-values will not be correct. The threshold for testing the first correlation should also be converted to a Z-score and read out from the graph in the same way. This transformation will likely increase the differences between conditions, and needs to be done.

Fisher's $Z=0.5*\ln((1+r)/(1-r))$

(the wikipedia page is my reference for this!
https://en.wikipedia.org/wiki/Fisher_transformation)

Nick Holmes

Decision letter (RSOS-172175.R0)

20-Feb-2018

Dear Dr Bergmann Tiest,

The editors assigned to your paper ("The influence of visual and haptic material information on early grasping force") have now received comments from reviewers. We would like you to revise your paper in accordance with the referee and Associate Editor suggestions which can be found below (not including confidential reports to the Editor). Please note this decision does not guarantee eventual acceptance.

Please submit a copy of your revised paper within three weeks (i.e. by the 15-Mar-2018). If we do not hear from you within this time then it will be assumed that the paper has been withdrawn. In exceptional circumstances, extensions may be possible if agreed with the Editorial Office in advance. We do not allow multiple rounds of revision so we urge you to make every effort to fully address all of the comments at this stage. If deemed necessary by the Editors, your manuscript will be sent back to one or more of the original reviewers for assessment. If the original reviewers are not available, we may invite new reviewers.

- Data accessibility

It is a condition of publication that all supporting data are made available either as supplementary information or preferably in a suitable permanent repository. The data accessibility section should state where the article's supporting data can be accessed. This section

should also include details, where possible of where to access other relevant research materials such as statistical tools, protocols, software etc can be accessed. If the data have been deposited in an external repository this section should list the database, accession number and link to the DOI for all data from the article that have been made publicly available. Data sets that have been deposited in an external repository and have a DOI should also be appropriately cited in the manuscript and included in the reference list.

If you wish to submit your supporting data or code to Dryad (<http://datadryad.org/>), or modify your current submission to dryad, please use the following link:
<http://datadryad.org/submit?journalID=RSOS&manu=RSOS-172175>

- **Competing interests**

- **Authors' contributions**

- **Acknowledgements**

- **Funding statement**

Please note that Royal Society Open Science will introduce article processing charges for all new submissions received from 1 January 2018. Charges will also apply to papers transferred to Royal Society Open Science from other Royal Society Publishing journals, as well as papers submitted as part of our collaboration with the Royal Society of Chemistry (<http://rsos.royalsocietypublishing.org/chemistry>). If your manuscript is submitted and accepted for publication after 1 Jan 2018, you will be asked to pay the article processing charge, unless you request a waiver and this is approved by Royal Society Publishing. You can find out more about the charges at <http://rsos.royalsocietypublishing.org/page/charges>. Should you have any queries, please contact openscience@royalsociety.org.

on behalf of Dr Stephen Jackson (Associate Editor) and Antonia Hamilton (Subject Editor)
openscience@royalsociety.org

Comments to Author:

Reviewers' Comments to Author:

Reviewer: 1

Comments to the Author(s)

This short paper reports two experiments – one addressing the static friction of a set of different objects, when shear force is applied to the point of slip by a human thumb – and the second comparing grip forces when participants grip and lift these objects with and without vision. The main result of the paper is to report differences in the impact forces (which I think is a better description of what the authors measure) and of grip/load force ratios after lift-off.

The paper follows in the line of previous work on this topic, but I think the question asked here (the effect of visual vs non-visual lifting, without in the latter case prior knowledge of the object), is novel. However, there are issues with the design that limit my enthusiasm.

First, measures of static friction are made by one of the experimenters. The authors report how this measure is influenced by e.g. skin moisture etc and therefore also by individual characteristics of the participants' fingers. Thus there may be differences in individuals' friction coefficients that might be important. It is a shame they did not measure this for each participant. The authors claim that the rank order of the friction estimate is unlikely to differ. This is somewhat undermined by their own data, showing for example that the variation in friction estimates (across only 2 measures) is as large as the difference in average friction between surfaces. Thus the rank order could well differ – in fact it does differ even between two measurements by the same person.

Second, they have no data on load force prior to the object lift-off. This is a critical phase of the grasping action. By and large, once lift off has occurred, the grip/load force ratio is sufficiently large that there is minimal difference across surface materials. However it seems quite likely that the participants, especially when deprived of vision, might modulate grip prior to lift-off, by sensing slip. This would be unmeasured in their data.

Third, although a minor point, the load forces are estimated by integration of positional data sampled at 100 Hz into acceleration estimates, and this is likely to be noisy. It may also be important that there is no measure of object rotation, and one would like to be able to rule out any rotational slips, again, especially in the non-vision conditions. It would be interesting also to report more directly on the load force, or object motion (eg acceleration), rather than only showing grip/load ratios or statistics of load force differences.

Finally, a lot of the paper is devoted to measures of “grip” force 10 ms after contact. I am sure that the force will be changing rapidly at this early moment, as it will be affected by the impact force (depending on speed of finger contact) and by object compliance, as the authors do acknowledge. While I accept that there is a danger of confounding later measurements by feedback responses, I think that there could easily be changes in “grip” for 10's of ms, driven only by these impact effects.

Minor comments

Please justify the use of the reciprocal of CoF rather than minus-CoF (if a positive correlation is necessary – why? – it would sufficient to invert the sign).

Please explain what complementary error function is (equation on page 6).

Reviewer: 2

Comments to the Author(s)

The authors have presented an excellently-written manuscript. I could not find a single typo! More importantly, the background, methods, results, and conclusions are all very clear and easy to read. Probably the best-written manuscript I've reviewed in several years. I am not an expert in this field, but I learnt a lot from this very-clearly written manuscript.

I only one minor comment, which refers to the parametric analysis of correlation coefficients. I'm sure the proposed changes will improve the manuscript, and make the between-condition differences stronger, but some of the conclusions about the timing may change a bit:

Minor comments

Page 7, line 42 - R-values need degrees of freedom - 4?

It would be better, in my opinion, here to calculate an r-value for each participant, to convert all the participants' r-values to Z-values (Fishers r-to-Z), then run a t-test on the Z scores, testing against zero. It is possible (though unlikely) that a correlation across (the means across participants for each of) the materials could arise, but that each participant does not show this correlation individually, and the group mean correlation is not significantly greater than zero. Better to test the within-participant correlation and express with confidence interval/standard error (on the Z-scores). Mean Z-scores could then be converted back into a group r-value if needed.

Figure 10 & statistics on page 8 - r-values need to be converted to Z values before any analysis: r is non-linear (and non-parametric, with bounds at -1 and 1), so the standard error bars are not representative and suffer from ceiling and floor effects, and the p-values will not be correct. The threshold for testing the first correlation should also be converted to a Z-score and read out from the graph in the same way. This transformation will likely increase the differences between conditions, and needs to be done.

Fisher's $Z=0.5*\ln((1+r)/(1-r))$

(the wikipedia page is my reference for this!

https://en.wikipedia.org/wiki/Fisher_transformation)

Nick Holmes

Author's Response to Decision Letter for (RSOS-172175.R0)

See Appendix A.

RSOS-172175.R1 (Revision)

Review form: Reviewer 3

Is the manuscript scientifically sound in its present form?

Yes

Are the interpretations and conclusions justified by the results?

Yes

Is the language acceptable?

Yes

Is it clear how to access all supporting data?

Yes

Do you have any ethical concerns with this paper?

No

Have you any concerns about statistical analyses in this paper?

No

Recommendation?

Accept with minor revision (please list in comments)

Comments to the Author(s)

The study by Bergmann Tiest et al. presents findings on the role of vision in the adaptation of fingertip forces to various surfaces with different frictional properties. The purpose of the study is to find out whether or not the people scale their fingertip grip forces based on visual information or not. To be able to answer their questions they conducted two experiments. First, the static coefficient of friction was measured using a tribometer. Then, they recorded the grip force while 12 subjects grasp and lift cubes coated with different surfaces with and without vision. They analyzed outcome measures like grip force/load force, maximum grip force, maximum load force, grip force 10 ms after onset, maximum grip/load forces. For their small sample set, they found no significant correlation between grip force and CoF in the first 100 ms after the grip force onset. It was concluded that the main cause for the anticipatory modulation of grip force is the haptic sensation of the texture, rather than vision, as there is no clear indication of the use of visual information to anticipate the friction and adjust grip forces.

Overall, the manuscript is well written; clear and logical and the research question is novel and interesting. It provides insight on the role of vision in haptic exploration. The literature review is substantial relative to the length of the manuscript. I have a couple criticisms regarding the methods and the experimental design as following:

1-I find the procedure for measuring the static friction unclear. Either a better description or an image with labeling sensors mentioned would make it clearer.

2-First, I agree with reviewer-1 regarding the critique about the CoF measured for all subjects would have made this data much stronger, authors mention "However, that would make it impossible to pool data per material, greatly hampering the statistical analysis" at line 24, but this could have been resolved by aggregating the CoF (averaging all subjects) per sample, which can give an idea about how susceptible some of these textures are to moisture or differences in skin structure. But I still find it acceptable to work with a single set measured with the experimenter's finger, as averaging across subject can cause a decrease in signal-to-noise ratio as well. However, I find the number of repeats very low given the variability in grip forces during tasks like these, ideally, they should measure the CoF with a confidence interval. This should be noted as a limitation.

3-I appreciate the due diligence the authors had with the calibration, considering the poor accuracy and repeatability of FSR sensors, and found the calibration section well written and useful. In my opinion, these sensors aren't appropriate to study grip forces precisely. They are known for having poor accuracy and used for qualitative results more than quantitative. I suggest that the authors include sensor characteristics in their methods (accuracy, repeatability).

4- From the motor control perspective, the anticipatory control of grip forces has two main components feedforward and feedback mechanisms. The first involves retrieving the friction information from memory and perhaps the vision to predict and apply the forces needed to grasp and lift the given object. This adjustment is done not only by controlling the amplitude but also the timing/speed of the change (i.e peak grip force/load force rate, time to peak grip/load force rates). In this study, the authors mainly focused on the amplitude component and showed that vision doesn't have an effect on the amplitude level during the anticipation of the friction. But more thorough analysis of the grip/load forces might be needed to make a more general conclusion. This can be also noted as a limitation.

Review form: Reviewer 4 (Gavin Buckingham)

Is the manuscript scientifically sound in its present form?

Yes

Are the interpretations and conclusions justified by the results?

Yes

Is the language acceptable?

Yes

Is it clear how to access all supporting data?

No

Do you have any ethical concerns with this paper?

No

Have you any concerns about statistical analyses in this paper?

No

Recommendation?

Major revision is needed (please make suggestions in comments)

Comments to the Author(s)

Apologies for the delay with this, and thanks for allowing me the extra time to review this work, which I read with great interest. I deliberately opted to review this 'blind', without reading the other reviewers' comments first.

First, although I liked this manuscript and the study it presents, I have some reservations about accepting it for publication. My comments on each section are outlined below, but I don't feel there is anything there which the authors cannot address. The journal does require data and analysis code be made available – perhaps I missed it but I didn't find any link to these materials in the manuscript.

Abstract:

The sentence starting "already at..." is very difficult to follow. Given how critical it is, I'd recommend re-phrasing

Introduction:

The sentence starting "Thus, people react..." doesn't make sense to me. Perhaps 'react to having'?

"this effect was termed the material-weight illusion" – the authors here are presumably referring to the way in which material cues drive grip and load force rates, rather than the perceptual illusion itself (the fact that this effect is demonstrated in the illusion-inducing stimuli is just an experimental nicety, and thus somewhat irrelevant to the point the authors are trying to make). I'd suggest re-phrasing to something more general like 'sensorimotor prediction driven by visual material cues'

Although the authors use the 2009 paper as a justification to miss out the first trial as a way to ignore the possible confound of material cues on object lifting behaviour, a follow up study did note that these material-induced prediction effects do seem to persist rather longer during a no-vision condition (Buckingham, G. Ranger, N.S., & Goodale, M.A. (2011). The material-weight illusion induced by expectations alone. *Attention, Perception, & Psychophysics*, 73, 36-41.). This seems like it requires some consideration, as this effect of removing vision on sensorimotor prediction has also been shown for size cues (Buckingham, G. Ranger, N.S., & Goodale, M.A. (2011). The role of vision in detecting and correcting fingertip force errors during object lifting. *Journal of Vision*, 11: doi: 10.1167/11.1.4.) Regardless, I'd have thought that these effects could be examined in the current dataset to determine how much data needs to be eliminated (not that I'm necessarily advocating that approach)

Materials and Methods

The CoF measurement seems a little unusual – it seems like it would be difficult to apply tangential force to a surface without any orthogonal force simply due to deformation of the finger pad. Isn't the standard procedure for this to grip the object firmly, and then slowly release the fingers, loosening the grip until slippage occurs?

3.1.2 – the point about the block being quite heavy wasn't quite clear to me – is the idea that because they blocks are heavier than would be expected, a underestimation of force would be used? Presumably though that would affect only the first few lifts (as per the intro). But perhaps I'm just misunderstanding.

I'd like some more details about the lift – was it a movement of the shoulder, elbow, or wrist? The latter surely would have significant rotational forces

Similarly, I'd like more details about the practice trials. What does 'from the same set' mean?

Were test objects themselves used as practice trials? If so, this needs some further discussion given the profound effects of prior experience and trial n-1 forces can have on the behaviour of

the grip/load force system

3.1.5 – again, here, I'd recommend the authors come up with a more appropriate term than the 'material-weight illusion', which has nothing to do with forces per se

Figure 7 – this is a great figure, but it would be useful to signal where in these ratio plots the timepoints for the less-obvious DVs (peak grip force and peak load force) occur

Results

Please include the p value for the non-significant 10ms peak grip force at 10ms (and other such non-significant results throughout)

Discussion

The word 'participant' is misspelled (about line 30, left column). The effect outlined in this paragraph does seem to suggest a rather sophisticated mechanism at play though – maybe the authors should discuss this in the context of (Mawase, & Karniel, 2010, Evidence for predictive control in lifting series of virtual objects *Experimental brain research* 203 (2), 447-452

The time course of this effect is similar to that reported in a recent paper using haptic cues to guide action which maybe deserves some discussion in this context (Pruszynski, J.A., Johansson, R.S., Flanagan, J.R. (2016). A rapid tactile-motor reflex automatically guides reaching towards handheld objects. *Current Biology* 26: 788-792.)

A further alternate explanation for the interaction with visual condition, which the authors perhaps don't fully acknowledge, is simply order effects – participants have had far more practice by the time they interact with visual cues.

Review form: Reviewer 5

Is the manuscript scientifically sound in its present form?

No

Are the interpretations and conclusions justified by the results?

Yes

Is the language acceptable?

Yes

Is it clear how to access all supporting data?

Yes

Do you have any ethical concerns with this paper?

No

Have you any concerns about statistical analyses in this paper?

No

Recommendation?

Major revision is needed (please make suggestions in comments)

Comments to the Author(s)

The manuscript by Bergmann Tiest and Kappers is well written and the topic is interesting and not well studied yet. However, I have a few comments that, when addressed, hopefully improve the manuscript:

1. Most importantly, I agree with reviewer 1 that the friction measurement by a single author is not sufficient. Friction is not the property of an object it depends on TWO surfaces, here the finger and the material. Thus, the friction will also be determined by the surface properties of the hand (e.g. the moisture) and as the authors state themselves the relation between the two is not linear. One might assume that for some materials, such as sandpaper, the friction coefficient will always be high (because the material is designed in that way), but without measurements we don't know this for sure and especially not for the other materials. Furthermore, the current measurements show large variability between repetitions (as also pointed out by reviewer 1) and they were conducted by the experimenter, not a naïve person. Because there is only data from one person, we have no idea about the interindividual differences or how friction might change over the course of an experiment.

In my opinion, the authors should either measure friction for each participant individually (and then use the rank order of friction coefficients for their stats or simply look at correlations) or restrain from the analyses related to friction. However, this would cut a substantial and interesting part of the manuscript. The remaining results, except analysis of load force (and comparison to grip force) were already presented In the author's 2014 conference paper. Individually determined friction coefficients would clearly make the results and conclusions of the paper much stronger.

2. The authors state that they cannot determine load force prior to movement onset, because they use position data of the infrared marker to do so. Additionally they could use data from the force sensor under the platform to determine load force until lift-off. This may improve the manuscript.

3. Determination of lift-off: This should be described more clearly. Was it only based on the platform sensor? If yes, what was the criterion? Or was this determined for each signal separately? Were the different measurements (force sensors, position data) not in sync and triggered at the same time?

4. The authors should give more details about the practice trials, how many were they? Vision or no vision condition etc.

5. What is the difference between metallic-like and wood-like plastic film? They seem to be quite similar. Do the films have any properties similar to wood and metal? The way the materials are described and named throughout the manuscript and in the figures suggests that they have something in common with real wood and real metal. It should be made clear that those are two different types of plastic (or three counting the acrylic surface).

6. The supplementary material should be labeled and explained more clearly. In its current form, it is not possible to understand the files and their content. A proper description of which file is which and what the different columns and rows stand for etc. would be helpful.

Decision letter (RSOS-172175.R1)

27-Jun-2018

Dear Dr Bergmann Tiest:

Manuscript ID RSOS-172175.R1 entitled "The influence of visual and haptic material information

on early grasping force" which you submitted to Royal Society Open Science, has been reviewed. The comments from reviewer(s) are included at the bottom of this letter.

In view of the criticisms of the reviewer(s), I must decline the manuscript for publication in Royal Society Open Science at this time. However, a new manuscript may be submitted which takes into consideration these comments.

Please note that resubmitting your manuscript does not guarantee eventual acceptance, and that your resubmission will be subject to re-review by the reviewer(s) before a decision is rendered.

You will be unable to make your revisions on the originally submitted version of your manuscript. Instead, revise your manuscript using a word processing program and save it on your computer.

You may also click the below link to start the resubmission process (or continue the process if you have already started your resubmission) for your manuscript. If you use the below link you will not be required to login to ScholarOne Manuscripts.

*** PLEASE NOTE: This is a two-step process. After clicking on the link, you will be directed to a webpage to confirm. ***

https://mc.manuscriptcentral.com/rsos?URL_MASK=ec7f3b93fd47481fa8bb0a371b42ca6b

Because we are trying to facilitate timely publication of manuscripts submitted to Royal Society Open Science, your resubmitted manuscript should be submitted by 25-Dec-2018. If you are unable to submit by this date please contact the Editorial Office for options.

Please note that Royal Society Open Science will introduce article processing charges for all new submissions received from 1 January 2018. Charges will also apply to papers transferred to Royal Society Open Science from other Royal Society Publishing journals, as well as papers submitted as part of our collaboration with the Royal Society of Chemistry (<http://rsos.royalsocietypublishing.org/chemistry>). If your manuscript is submitted and accepted for publication after 1 Jan 2018, you will be asked to pay the article processing charge, unless you request a waiver and this is approved by Royal Society Publishing. You can find out more about the charges at <http://rsos.royalsocietypublishing.org/page/charges>. Should you have any queries, please contact openscience@royalsociety.org.

I look forward to a resubmission.

Kind regards,
Royal Society Open Science Editorial Office
openscience@royalsociety.org

on behalf of Dr Stephen Jackson (Associate Editor) and Prof. Antonia Hamilton (Subject Editor)
openscience@royalsociety.org

Reviewer comments to Author:

Reviewer: 3

Comments to the Author(s)

The study by Bergmann Tiest et al. presents findings on the role of vision in the adaptation of fingertip forces to various surfaces with different frictional properties. The purpose of the study is to find out whether or not the people scale their fingertip grip forces based on visual information or not. To be able to answer their questions they conducted two experiments. First, the static coefficient of friction was measured using a tribometer. Then, they recorded the grip force while 12 subjects grasp and lift cubes coated with different surfaces with and without vision. They analyzed outcome measures like grip force/load force, maximum grip force, maximum load force, grip force 10 ms after onset, maximum grip/load forces. For their small sample set, they found no significant correlation between grip force and CoF in the first 100 ms after the grip force onset. It was concluded that the main cause for the anticipatory modulation of grip force is the haptic sensation of the texture, rather than vision, as there is no clear indication of the use of visual information to anticipate the friction and adjust grip forces.

Overall, the manuscript is well written; clear and logical and the research question is novel and interesting. It provides insight on the role of vision in haptic exploration. The literature review is substantial relative to the length of the manuscript. I have a couple criticisms regarding the methods and the experimental design as following:

1-I find the procedure for measuring the static friction unclear. Either a better description or an image with labeling sensors mentioned would make it clearer.

2-First, I agree with reviewer-1 regarding the critique about the CoF measured for all subjects would have made this data much stronger, authors mention "However, that would make it impossible to pool data per material, greatly hampering the statistical analysis" at line 24, but this could have been resolved by aggregating the CoF (averaging all subjects) per sample, which can give an idea about how susceptible some of these textures are to moisture or differences in skin structure. But I still find it acceptable to work with a single set measured with the experimenter's finger, as averaging across subject can cause a decrease in signal-to-noise ratio as well. However, I find the number of repeats very low given the variability in grip forces during tasks like these, ideally, they should measure the CoF with a confidence interval. This should be noted as a limitation.

3-I appreciate the due diligence the authors had with the calibration, considering the poor accuracy and repeatability of FSR sensors, and found the calibration section well written and useful. In my opinion, these sensors aren't appropriate to study grip forces precisely. They are known for having poor accuracy and used for qualitative results more than quantitative. I suggest that the authors include sensor characteristics in their methods (accuracy, repeatability).

4- From the motor control perspective, the anticipatory control of grip forces has two main components feedforward and feedback mechanisms. The first involves retrieving the friction information from memory and perhaps the vision to predict and apply the forces needed to grasp and lift the given object. This adjustment is done not only by controlling the amplitude but also the timing/speed of the change (i.e peak grip force/load force rate, time to peak grip/load force rates). In this study, the authors mainly focused on the amplitude component and showed that vision doesn't have an effect on the amplitude level during the anticipation of the friction. But more thorough analysis of the grip/load forces might be needed to make a more general conclusion. This can be also noted as a limitation.

Reviewer: 4

Comments to the Author(s)

Apologies for the delay with this, and thanks for allowing me the extra time to review this work, which I read with great interest. I deliberately opted to review this 'blind', without reading the other reviewers' comments first.

First, although I liked this manuscript and the study it presents, I have some reservations about accepting it for publication. My comments on each section are outlined below, but I don't feel there is anything there which the authors cannot address. The journal does require data and analysis code be made available – perhaps I missed it but I didn't find any link to these materials in the manuscript.

Abstract:

The sentence starting "already at..." is very difficult to follow. Given how critical it is, I'd recommend re-phrasing

Introduction:

The sentence starting "Thus, people react..." doesn't make sense to me. Perhaps 'react to having'?

"this effect was termed the material-weight illusion" – the authors here are presumably referring to the way in which material cues drive grip and load force rates, rather than the perceptual illusion itself (the fact that this effect is demonstrated in the illusion-inducing stimuli is just an experimental nicety, and thus somewhat irrelevant to the point the authors are trying to make). I'd suggest re-phrasing to something more general like 'sensorimotor prediction driven by visual material cues'

Although the authors use the 2009 paper as a justification to miss out the first trial as a way to ignore the possible confound of material cues on object lifting behaviour, a follow up study did note that these material-induced prediction effects do seem to persist rather longer during a no-vision condition (Buckingham, G. Ranger, N.S., & Goodale, M.A. (2011). The material-weight illusion induced by expectations alone. *Attention, Perception, & Psychophysics*, 73, 36-41.). This seems like it requires some consideration, as this effect of removing vision on sensorimotor prediction has also been shown for size cues (Buckingham, G. Ranger, N.S., & Goodale, M.A. (2011). The role of vision in detecting and correcting fingertip force errors during object lifting. *Journal of Vision*, 11: doi: 10.1167/11.1.4.) Regardless, I'd have thought that these effects could be examined in the current dataset to determine how much data needs to be eliminated (not that I'm necessarily advocating that approach)

Materials and Methods

The CoF measurement seems a little unusual – it seems like it would be difficult to apply tangential force to a surface without any orthogonal force simply due to deformation of the finger pad. Isn't the standard procedure for this to grip the object firmly, and then slowly release the fingers, loosening the grip until slippage occurs?

3.1.2 – the point about the block being quite heavy wasn't quite clear to me – is the idea that because they blocks are heavier than would be expected, a underestimation of force would be used? Presumably though that would affect only the first few lifts (as per the intro). But perhaps I'm just misunderstanding.

I'd like some more details about the lift – was it a movement of the shoulder, elbow, or wrist? The latter surely would have significant rotational forces

Similarly, I'd like more details about the practice trials. What does 'from the same set' mean? Were test objects themselves used as practice trials? If so, this needs some further discussion

given the profound effects of prior experience and trial n-1 forces can have on the behaviour of the grip/load force system

3.1.5 – again, here, I'd recommend the authors come up with a more appropriate term than the 'material-weight illusion', which has nothing to do with forces per se

Figure 7 – this is a great figure, but it would be useful to signal where in these ratio plots the timepoints for the less-obvious DVs (peak grip force and peak load force) occur

Results

Please include the p value for the non-significant 10ms peak grip force at 10ms (and other such non-significant results throughout)

Discussion

The word 'participant' is misspelled (about line 30, left column). The effect outlined in this paragraph does seem to suggest a rather sophisticated mechanism at play though – maybe the authors should discuss this in the context of (Mawase, & Karniel, 2010, Evidence for predictive control in lifting series of virtual objects *Experimental brain research* 203 (2), 447-452

The time course of this effect is similar to that reported in a recent paper using haptic cues to guide action which maybe deserves some discussion in this context (Pruszynski, J.A., Johansson, R.S., Flanagan, J.R. (2016). A rapid tactile-motor reflex automatically guides reaching towards handheld objects. *Current Biology* 26: 788-792.)

A further alternate explanation for the interaction with visual condition, which the authors perhaps don't fully acknowledge, is simply order effects – participants have had far more practice by the time they interact with visual cues.

Reviewer: 5

Comments to the Author(s)

The manuscript by Bergmann Tiest and Kappers is well written and the topic is interesting and not well studied yet. However, I have a few comments that, when addressed, hopefully improve the manuscript:

1. Most importantly, I agree with reviewer 1 that the friction measurement by a single author is not sufficient. Friction is not the property of an object it depends on TWO surfaces, here the finger and the material. Thus, the friction will also be determined by the surface properties of the hand (e.g. the moisture) and as the authors state themselves the relation between the two is not linear. One might assume that for some materials, such as sandpaper, the friction coefficient will always be high (because the material is designed in that way), but without measurements we don't know this for sure and especially not for the other materials. Furthermore, the current measurements show large variability between repetitions (as also pointed out by reviewer 1) and they were conducted by the experimenter, not a naïve person. Because there is only data from one person, we have no idea about the interindividual differences or how friction might change over the course of an experiment.

In my opinion, the authors should either measure friction for each participant individually (and then use the rank order of friction coefficients for their stats or simply look at correlations) or restrain from the analyses related to friction. However, this would cut a substantial and interesting part of the manuscript. The remaining results, except analysis of load force (and comparison to grip force) were already presented in the author's 2014 conference paper. Individually determined friction coefficients would clearly make the results and conclusions of the paper much stronger.

2. The authors state that they cannot determine load force prior to movement onset, because they use position data of the infrared marker to do so. Additionally they could use data from the force sensor under the platform to determine load force until lift-off. This may improve the manuscript.
3. Determination of lift-off: This should be described more clearly. Was it only based on the platform sensor? If yes, what was the criterion? Or was this determined for each signal separately? Were the different measurements (force sensors, position data) not in sync and triggered at the same time?
4. The authors should give more details about the practice trails, how many were they? Vision or no vision condition etc.
5. What is the difference between metallic-like and wood-like plastic film? They seem to be quite similar. Do the films have any properties similar to wood and metal? The way the materials are described and named throughout the manuscript and in the figures suggests that they have something in common with real wood and real metal. It should be made clear that those are two different types of plastic (or three counting the acrylic surface).
6. The supplementary material should be labeled and explained more clearly. In its current form, it is not possible to understand the files and their content. A proper description of which file is which and what the different columns and rows stand for etc. would be helpful.

Author's Response to Decision Letter for (RSOS-172175.R1)

See Appendix B.

RSOS-181563.R0

Review form: Reviewer 3

Is the manuscript scientifically sound in its present form?

Yes

Are the interpretations and conclusions justified by the results?

Yes

Is the language acceptable?

Yes

Is it clear how to access all supporting data?

Yes

Do you have any ethical concerns with this paper?

No

Have you any concerns about statistical analyses in this paper?

No

Recommendation?

Accept as is

Comments to the Author(s)

Thank you for addressing my points.

Review form: Reviewer 4 (Gavin Buckingham)

Is the manuscript scientifically sound in its present form?

Yes

Are the interpretations and conclusions justified by the results?

Yes

Is the language acceptable?

Yes

Is it clear how to access all supporting data?

Yes

Do you have any ethical concerns with this paper?

No

Have you any concerns about statistical analyses in this paper?

No

Recommendation?

Accept as is

Comments to the Author(s)

The authors have done a fine job of addressing my comments. My one issue, which can be addressed at the proofs stage, is the use of the phrase 'over-dimensioned' in the clarification added to 3.1.2 - this isn't a term which really makes any sense to me. 'Larger-than-necessary' perhaps?

Review form: Reviewer 5

Is the manuscript scientifically sound in its present form?

Yes

Are the interpretations and conclusions justified by the results?

Yes

Is the language acceptable?

Yes

Is it clear how to access all supporting data?

Yes

Do you have any ethical concerns with this paper?

No

Have you any concerns about statistical analyses in this paper?

No

Recommendation?

Major revision is needed (please make suggestions in comments)

Comments to the Author(s)

The authors addressed most of my concerns. However, my main concern is remaining: The fact that friction measurements were only conducted by one of the authors and not every participant.

I disagree with the arguments of the authors:

1) It is not impossible to add friction measurements of individual participants. To me it sounds as if the authors took the data from their 2014 conference proceeding and added the friction measurements by one of the authors post-hoc to do an additional (and in principle very interesting) analysis. While I don't generally disagree with that approach and some of the other new analyses are well done, I am still convinced that friction should be measured individually. Again, friction is a property of two surfaces. Thus, if one wants to test the effect of friction, one has to measure friction. If it is not possible to do this for the same participants, that may just mean collecting data of new participants.

2) I also don't agree with their statistical argument. First, currently the author assume that there are no large differences between friction coefficients of different participants (otherwise they would have to measure it individually), why are they then so worried that they could not pool data of different participants across different materials? Even if friction coefficients are not exactly the same, they could be binned or rank-ordered for that specific analysis. After all we would at least know whether (and how much) they differ between persons. Conversely, I think individual measurements would strengthen their argument enormously and some of their analyses are well possible with individual data (e.g. linear fit in figure 9) and would show the effect of friction on grip force very convincingly.

I think at the very least the authors should measure the friction coefficients for more (naïve) individuals, so that we are able to assess the interindividual variability for these materials.

Additionally the should present data from the literature to support their underlying assumption that the interindividual differences are smaller than the inter-material differences.

Other comments:

1) How can the lift-off be determined by using a threshold for the sensor underneath the platform, if that sensor was not calibrated?

Decision letter (RSOS-181563.R0)

21-Nov-2018

Dear Dr Bergmann Tiest,

The Subject Editor assigned to your paper ("The influence of visual and haptic material information on early grasping force") has now received comments from reviewers. We would like you to revise your paper in accordance with the referee and Associate Editor suggestions which can be found below (not including confidential reports to the Editor). Please note this decision does not guarantee eventual acceptance.

Please submit a copy of your revised paper before 14-Dec-2018. Please note that the revision deadline will expire at 00.00am on this date. If we do not hear from you within this time then it will be assumed that the paper has been withdrawn. In exceptional circumstances, extensions may be possible if agreed with the Editorial Office in advance. We do not allow multiple rounds of revision so we urge you to make every effort to fully address all of the comments at this stage. If deemed necessary by the Editors, your manuscript will be sent back to one or more of the original reviewers for assessment. If the original reviewers are not available we may invite new reviewers.

When submitting your revised manuscript, you must respond to the comments made by the referees and upload a file "Response to Referees" in "Section 6 - File Upload". Please use this to document how you have responded to each of the comments, and the adjustments you have made. In order to expedite the processing of the revised manuscript, please be as specific as possible in your response.

- Ethics statement

- Data accessibility

If you wish to submit your supporting data or code to Dryad (<http://datadryad.org/>), or modify your current submission to dryad, please use the following link:
<http://datadryad.org/submit?journalID=RSOS&manu=RSOS-181563>

- Competing interests

- Authors' contributions

- Acknowledgements

- Funding statement

on behalf of Dr Stephen Jackson (Associate Editor) and Antonia Hamilton (Subject Editor)
openscience@royalsociety.org

Subject Editor comments -

reviewer 5 has several comments, but it may be simplest to deal with the issue of friction by following the recommendation to "measure the friction coefficients for more (naïve) individuals, so that we are able to assess the interindividual variability for these materials". Collecting more data beyond that is possible but not necessary.

Reviewer comments to Author:

Reviewer: 4

Comments to the Author(s)

The authors have done a fine job of addressing my comments. My one issue, which can be addressed at the proofs stage, is the use of the phrase 'over-dimensioned' in the clarification added to 3.1.2 - this isn't a term which really makes any sense to me. 'Larger-than-necessary' perhaps?

Reviewer: 5

Comments to the Author(s)

The authors addressed most of my concerns. However, my main concern is remaining: The fact that friction measurements were only conducted by one of the authors and not every participant. I disagree with the arguments of the authors:

1) It is not impossible to add friction measurements of individual participants. To me it sounds as if the authors took the data from their 2014 conference proceeding and added the friction measurements by one of the authors post-hoc to do an additional (and in principle very interesting) analysis. While I don't generally disagree with that approach and some of the other new analyses are well done, I am still convinced that friction should be measured individually. Again, friction is a property of two surfaces. Thus, if one wants to test the effect of friction, one has to measure friction. If it is not possible to do this for the same participants, that may just mean collecting data of new participants.

2) I also don't agree with their statistical argument. First, currently the author assume that there are no large differences between friction coefficients of different participants (otherwise they would have to measure it individually), why are they then so worried that they could not pool data of different participants across different materials? Even if friction coefficients are not exactly the same, they could be binned or rank-ordered for that specific analysis. After all we would at least know whether (and how much) they differ between persons. Conversely, I think individual measurements would strengthen their argument enormously and some of their analyses are well possible with individual data (e.g. linear fit in figure 9) and would show the effect of friction on grip force very convincingly.

I think at the very least the authors should measure the friction coefficients for more (naïve) individuals, so that we are able to assess the interindividual variability for these materials. Additionally the should present data from the literature to support their underlying assumption that the interindividual differences are smaller than the inter-material differences.

Other comments:

1) How can the lift-off be determined by using a threshold for the sensor underneath the platform, if that sensor was not calibrated?

Reviewer: 3

Comments to the Author(s)

Thank you for addressing my points.

Author's Response to Decision Letter for (RSOS-181563.R0)

See Appendix C.

Decision letter (RSOS-181563.R1)

01-Feb-2019

Dear Dr Bergmann Tiest,

I am pleased to inform you that your manuscript entitled "The influence of visual and haptic material information on early grasping force" is now accepted for publication in Royal Society Open Science.

on behalf of Dr Stephen Jackson (Associate Editor) and Antonia Hamilton (Subject Editor)
openscience@royalsociety.org

Follow Royal Society Publishing on Twitter: [@RSocPublishing](https://twitter.com/RSocPublishing)

Appendix A

Dear dr Jackson,

Please find below our response to the reviewers' comments and how the manuscript was changed as a result.

Reviewer 1

This short paper reports two experiments – one addressing the static friction of a set of different objects, when shear force is applied to the point of slip by a human thumb – and the second comparing grip forces when participants grip and lift these objects with and without vision. The main result of the paper is to report differences in the impact forces (which I think is a better description of what the authors measure) and of grip/load force ratios after lift-off. The paper follows in the line of previous work on this topic, but I think the question asked here (the effect of visual vs non-visual lifting, without in the latter case prior knowledge of the object), is novel. However, there are issues with the design that limit my enthusiasm.

First, measures of static friction are made by one of the experimenters. The authors report how this measure is influenced by e.g. skin moisture etc and therefore also by individual characteristics of the participants' fingers. Thus there may be differences in individuals' friction coefficients that might be important. It is a shame they did not measure this for each participant. The authors claim that the rank order of the friction estimate is unlikely to differ. This is somewhat undermined by their own data, showing for example that the variation in friction estimates (across only 2 measures) is as large as the difference in average friction between surfaces. Thus the rank order could well differ – in fact it does differ even between two measurements by the same person.

We agree that the reliability of the friction measurements may have been overstated. Therefore, we have removed the sentence “Furthermore, although absolute friction values might be different from person to person, the ordering of the stimuli according to friction is likely to be more or less the same for all participants”. On the other hand, the close relationship illustrated in Fig. 9 suggests that on average, the friction values measured with the experimenter are in accordance with the participants' perception, in the sense that on average, the participants use a higher maximum grip force for more slippery objects. This has reinforced our decision to use a single set of friction values, enabling us to pool the participants' data, strengthening the statistical analysis. This is now mentioned in section 3.2.

Second, they have no data on load force prior to the object lift-off. This is a critical phase of the grasping action. By and large, once lift off has occurred, the grip/load force ratio is sufficiently large that there is minimal difference across surface materials. However it seems quite likely that the participants,

especially when deprived of vision, might modulate grip prior to lift-off, by sensing slip. This would be unmeasured in their data.

Here, we only partially agree with the reviewer. Indeed, we do not have load force data prior to stimulus lift-off, but we do have grip force data. Thus, the modulation in grip force prior to lift-off is not “unmeasured” in our data, as the reviewer states, but is included in the data, even before lift-off. We have added the following text to the method section to explain this: “For this reason, we are unable to measure the load force prior to lift-off with this method. However, for our main analysis we are interested in the early grip force, which is available also before lift-off.”

Third, although a minor point, the load forces are estimated by integration of positional data sampled at 100 Hz into acceleration estimates, and this is likely to be noisy. It may also be important that there is no measure of object rotation, and one would like to be able to rule out any rotational slips, again, especially in the non-vision conditions. It would be interesting also to report more directly on the load force, or object motion (eg acceleration), rather than only showing grip/load ratios or statistics of load force differences.

Just to clarify: the load forces are not estimated by integration, but by differentiation of positional data, which is less prone to be noisy.

Regarding the object rotation, we expect rotational forces to be minimal since participants were instructed to perform a vertical lift. We have added the following text to section 3.1.2 to explain this: “Using this sensor, the grip force perpendicular to the surface could be measured. Rotational forces could not be measured, but since participants were instructed to perform a vertical lift, these will likely be insignificant, especially in the early phase of the lift.”

Regarding the load forces, we do not think that a more extensive report on the load forces will add much to the article, as all objects’ masses were equal, and thus also their gravitational and inertial forces. The same goes for the acceleration, which is directly related to the load force.

Finally, a lot of the paper is devoted to measures of “grip” force 10 ms after contact. I am sure that the force will be changing rapidly at this early moment, as it will be affected by the impact force (depending on speed of finger contact) and by object compliance, as the authors do acknowledge. While I accept that there is a danger of confounding later measurements by feedback responses, I think that there could easily be changes in “grip” for 10’s of ms, driven only by these impact effects.

This is an interesting point that we had not yet considered. It may be that the way the finger impacts on the surface causes differences in measured force. This possible explanation is now added to section 4.

Minor comments

Please justify the use of the reciprocal of CoF rather than minus-CoF (if a positive correlation is necessary – why? – it would sufficient to invert the sign).

Thank you for pointing this out; this was phrased unclearly. We meant to say “The reciprocal of the CoF was used, because then a linear relationship is expected.” This is an advantage, because a correlation analysis deals better with a linear relationship. This is now corrected in section 3.1.5.

Please explain what complementary error function is (equation on page 6).

We have added “i.e. the inverse of $(1 - \text{erf})$ ” to section 3.1.5.

Reviewer 2

The authors have presented an excellently-written manuscript. I could not find a single typo! More importantly, the background, methods, results, and conclusions are all very clear and easy to read. Probably the best-written manuscript I’ve reviewed in several years. I am not an expert in this field, but I learnt a lot from this very-clearly written manuscript.

I only one minor comment, which refers to the parametric analysis of correlation coefficients. I’m sure the proposed changes will improve the manuscript, and make the between-condition differences stronger, but some of the conclusions about the timing may change a bit:

Minor comments Page 7, line 42 - R-values need degrees of freedom - 4?

Fixed.

It would be better, in my opinion, here to calculate an r-value for each participant, to convert all the participants’ r-values to Z-values (Fishers r-to-Z), then run a t-test on the Z scores, testing against zero. It is possible (though unlikely) that a correlation across (the means across participants for each of) the materials could arise, but that each participant does not show this correlation individually, and the group mean correlation is not significantly greater than zero. Better to test the within-participant correlation and express with confidence interval/standard error (on the Z-scores). Mean Z-scores could then be converted back into a group r-value if needed.

We have followed the above procedure and explained this in section 3.1.5: “For each participant, the maximum grip forces averaged per material were correlated with the reciprocal measured friction coefficients. The correlation coefficients R were transformed to z -scores using Fisher’s transform and subsequently averaged. The average z -score was transformed back to an R -value”. The new R -values are now reported in section 3.2.

Figure 10 & statistics on page 8 - r-values need to be converted to Z values before any analysis: r is non-linear (and non-parametric, with bounds at -1 and 1), so the standard error bars are not representative and suffer from ceiling and floor effects, and the p-values will not be correct. The threshold for testing the first correlation should also be converted to a Z-score and read out from the graph in the same way. This transformation will likely increase the differences between conditions, and needs to be done.

We have adapted Fig. 10 according to the above procedure. The difference between the conditions has not increased.

Appendix B

Dear dr. Jackson, dear Stephen,

Thank your for handling our manuscript submitted to Royal Society Open Science and entitled "The influence of visual and haptic material information on early grasping force" (Manuscript ID RSOS-172175.R1).

To be honest, we were a bit surprised that after revising our previous manuscript and addressing all concerns and suggestions of the two reviewers (one of whom found this the best manuscript he reviewed in years), our manuscript was now assessed by three new reviewers. However, we think that by addressing the comments of these three new reviewers our manuscript again improved. We sincerely hope that you and the reviewers will be satisfied with the current version.

Below we copied and italicized all the comments of the reviewers. Below each comment, we explain in green how we dealt with the comment and which texts we added to our manuscript.

With kind regards,

Wouter M. Bergmann Tiest
Astrid M.L. Kappers

Reviewer comments to Author:

Reviewer: 3

Comments to the Author(s)

The study by Bergmann Tiest et al. presents findings on the role of vision in the adaptation of fingertip forces to various surfaces with different frictional properties. The purpose of the study is to find out whether or not the people scale their fingertip grip forces based on visual information or not. To be able to answer their questions they conducted two experiments. First, the static coefficient of friction was measured using a tribometer. Then, they recorded the grip force while 12 subjects grasp and lift cubes coated with different surfaces with and without vision. They analyzed outcome measures like grip force/load force, maximum grip force, maximum load force, grip force 10 ms after onset, maximum grip/load forces. For their small sample set, they found no significant correlation between grip force and CoF in the first 100 ms after the grip force onset. It was concluded that the main cause for the anticipatory modulation of grip force is the haptic sensation of the texture, rather

than vision, as there is no clear indication of the use of visual information to anticipate the friction and adjust grip forces.

Overall, the manuscript is well written; clear and logical and the research question is novel and interesting. It provides insight on the role of vision in haptic exploration. The literature review is substantial relative to the length of the manuscript.

Thank you for this positive assessment.

I have a couple criticisms regarding the methods and the experimental design as following:

1-I find the procedure for measuring the static friction unclear. Either a better description or an image with labeling sensors mentioned would make it clearer.

For the static friction measurements, we used a device and method similar to the one used by Platkiewicz et al. [22]. A figure of the set-up can be found in this reference. We added this reference to the text as follows:

- , we used a device described in Platkiewicz et al. [22].
- This method is similar to the one used by Platkiewicz et al. [22].
- [22] Platkiewicz J, Mansutti A, Bordegoni M, Hayward V. Recording Device for Natural Haptic Textures Felt with the Bare Fingertip. In: Auvray M, Duriez C, editors. Haptics: Neuroscience, Devices, Modeling, and Applications. Part I. vol. 8618 of Lecture Notes in Computer Science. Berlin/Heidelberg: Springer-Verlag; 2014. p. 521–528.

2-First, I agree with reviewer-1 regarding the critique about the CoF measured for all subjects would have made this data much stronger, authors mention “However, that would make it impossible to pool data per material, greatly hampering the statistical analysis” at line 24, but this could have been resolved by aggregating the CoF (averaging all subjects) per sample, which can give an idea about how susceptible some of these textures are to moisture or differences in skin structure. But I still find it acceptable to work with a single set measured with the experimenter’s finger, as averaging across subject can cause a decrease in signal-to-noise ratio as well. However, I find the number of repeats very low given the variability in grip forces during tasks like these, ideally, they should measure the CoF with a confidence interval. This should be noted as a limitation.

To address this limitation, we added the following text to the Discussion:

In the present study, CoFs were not measured for each participant separately, but only with the experimenter’s finger, so as to be able to pool the data per material. Another way to do this would have been to measure the friction per

participant separately, but average the data over participants. This was not done, as averaging across participants would cause a decrease in signal-to-noise ratio. As shown in table 1, only two CoF measurements were performed for each material. Although this is a limitation, this was deemed acceptable as the difference between the two measurements for each material is considerably smaller than the range of measured CoFs.

3-I appreciate the due diligence the authors had with the calibration, considering the poor accuracy and repeatability of FSR sensors, and found the calibration section well written and useful. In my opinion, these sensors aren't appropriate to study grip forces precisely. They are known for having poor accuracy and used for qualitative results more than quantitative. I suggest that the authors include sensor characteristics in their methods (accuracy, repeatability).

We added information about the accuracy and repeatability to subsection 3.1.2 Stimuli. We also give a reference to the factsheet of the manufacturer:

- accuracy 0.01 N, repeatability $\pm 2\%$ [23]
- [23] Interlink Electronics. FSR 400 Series Data Sheet; 2013. <https://www.interlinkelectronics.com/fsr-400>.

4- From the motor control perspective, the anticipatory control of grip forces has two main components feedforward and feedback mechanisms. The first involves retrieving the friction information from memory and perhaps the vision to predict and apply the forces needed to grasp and lift the given object. This adjustment is done not only by controlling the amplitude but also the timing/speed of the change (i.e peak grip force/load force rate, time to peak grip/load force rates). In this study, the authors mainly focused on the amplitude component and showed that vision doesn't have an effect on the amplitude level during the anticipation of the friction. But more thorough analysis of the grip/load forces might be needed to make a more general conclusion. This can be also noted as a limitation.

At the end of the Analysis section we added the following text:

The analysis in the present study focused mainly on the maximum grip force and its correlation with the CoF. Other characteristics, such as grip force rate, load force rate or time-to-peak are not considered. Analysis of these characteristics might yield other insights, but does not contribute to answering our main research question.

Reviewer: 4

Apologies for the delay with this, and thanks for allowing me the extra time to review this work, which I read with great interest. I deliberately opted to review this 'blind', without reading the other reviewers' comments first.

First, although I liked this manuscript and the study it presents, I have some reservations about accepting it for publication. My comments on each section are outlined below, but I don't feel there is anything there which the authors cannot address.

We did our best to address all your concerns.

The journal does require data and analysis code be made available – perhaps I missed it but I didn't find any link to these materials in the manuscript.

At the end of the manuscript, somewhere between Conclusion and References, there was already a subsection "Data, code and materials" where a link to all the materials was given. We checked the link and the data are indeed available:

**The dataset supporting this article are available at
<http://hdl.handle.net/10411/C5QQGF>.**

Abstract:

The sentence starting "already at..." is very difficult to follow. Given how critical it is, I'd recommend re-phrasing

We rephrased this sentence to:

Already at an early phase in the grasp (before lift-off), the grip force correlated highly with the textures' static coefficient of friction.

Introduction:

The sentence starting "Thus, people react..." doesn't make sense to me. Perhaps 'react to having'?

We agree. We added the word "having".

"this effect was termed the material-weight illusion" – the authors here are presumably referring to the way in which material cues drive grip and load force rates, rather than the perceptual illusion itself (the fact that this effect is demonstrated in the illusion-inducing stimuli is just an experimental nicety, and thus somewhat irrelevant to the point the authors are trying to make). I'd suggest re-phrasing to something more general like 'sensorimotor prediction driven by visual material cues'

We rephrased "This effect was termed the material-weight illusion, and might cause ..." to:

This effect of sensorimotor prediction driven by visual material cues might cause ...

and we changed the sentence beginning with "Fortunately, it was found..." to:

Fortunately, *although the perceptual effect persisted (termed material-weight-illusion), the sensorimotor effect disappeared: it was found...*

In the Discussion we added the following text (here in italics):

This might have been caused by remnants of *the sensorimotor effect associated with the material-weight-illusion, ...*

Although the authors use the 2009 paper as a justification to miss out the first trial as a way to ignore the possible confound of material cues on object lifting behaviour, a follow up study did note that these material-induced prediction effects do seem to persist rather longer during a no-vision condition (Buckingham, G. Ranger, N.S., & Goodale, M.A. (2011). The material-weight illusion induced by expectations alone. Attention, Perception, & Psychophysics, 73, 36-41.). This seems like it requires some consideration, as this effect of removing vision on sensorimotor prediction has also been shown for size cues (Buckingham, G. Ranger, N.S., & Goodale, M.A. (2011). The role of vision in detecting and correcting fingertip force errors during object lifting. Journal of Vision, 11: doi: 10.1167/11.1.4.) Regardless, I'd have thought that these effects could be examined in the current dataset to determine how much data needs to be eliminated (not that I'm necessarily advocating that approach)

In our experiment, the same stimulus set was also used for the practice trials, so in fact, participants were provided with more familiarity of the stimuli than just the first trials. A clearer description of the practice trials was added to 3.1.4 (see below, where we address the practice trials).

In the Discussion, we now refer to the first 2011-paper cited above:

This is in line with more recent findings by the same group, showing an effect of expected mass on maximum load force that remains significant for four lifts, compared to just one or two for the effect on maximum grip force [24].

To the References we added:

[24] Buckingham G, Ranger NS, Goodale MA. The material-weight illusion induced by expectations alone. Attention, Perception, & Psychophysics. 2011;73:36-41.

Moreover we would like to note that also in the no-vision condition in the paper cited above, the participants do see the stimulus until a certain moment. That is unlike in our experiment in which participants were blindfolded.

Materials and Methods

The CoF measurement seems a little unusual – it seems like it would be difficult to apply tangential force to a surface without any orthogonal force simply due to deformation of the finger pad. Isn't the standard procedure for this to grip the object firmly, and then slowly release the fingers, loosening the grip until slippage occurs?

Reviewer 3 also commented on our method for measuring friction (see comment 1). Here we repeat our response.

For the static friction measurements, we used a device and method similar to the one used by

Platkiewicz et al. [22]. A figure of the set-up can be found in this reference. We added this reference to the text as follows:

- , we used a device described in Platkiewicz et al. [22].
- This method is similar to the one used by Platkiewicz et al. [22].
- [22] Platkiewicz J, Mansutti A, Bordegoni M, Hayward V. Recording Device for Natural Haptic Textures Felt with the Bare Fingertip. In: Auvray M, Duriez C, editors. Haptics: Neuroscience, Devices, Modeling, and Applications. Part I. vol. 8618 of Lecture Notes in Computer Science. Berlin/Heidelberg: Springer-Verlag; 2014. p. 521–528.

3.1.2 – the point about the block being quite heavy wasn't quite clear to me – is the idea that because they blocks are heavier than would be expected, a underestimation of force would be used? Presumably though that would affect only the first few lifts (as per the intro). But perhaps I'm just misunderstanding.

We added the following clarification to 3.1.2:

With lighter blocks, participants might have been tempted to use the same, over-dimensioned grip force for all materials, ignoring material differences. As it was clear that the blocks were only covered by the different materials, and consisted internally of the same material, there was no suggestion that the blocks had different masses. For this reason, we expect no large effects of the material-weight-illusion [19].

I'd like some more details about the lift – was it a movement of the shoulder, elbow, or wrist? The latter surely would have significant rotational forces.

We added the following text to 3.1.4:

The elbow rested on the table during the whole lift, and rotational movement was kept to a minimum.

Similarly, I'd like more details about the practice trials. What does 'from the same set' mean? Were test objects themselves used as practice trials? If so, this needs

some further discussion given the profound effects of prior experience and trial n-1 forces can have on the behaviour of the grip/load force system.

We extended the description of the practice trials in 3.1.4:

Before the experiment proper began, the blindfolded participant was allowed one or two practice trials using the same stimuli as used in the experiment. Having been exposed to some of the stimuli before the experiment probably reduced the sensorimotor effect associated with the material-weight-illusion during the experiment.

3.1.5 – again, here, I'd recommend the authors come up with a more appropriate term than the 'material-weight illusion', which has nothing to do with forces per se

We rephrased the first sentence of 3.1.5 from "... to avoid effects of the material-weight-illusion" to:

... to avoid the sensorimotor effect associated with the material-weight-illusion.

Figure 7 – this is a great figure, but it would be useful to signal where in these ratio plots the timepoints for the less-obvious DVs (peak grip force and peak load force) occur

Thank you for this compliment!

We are sorry, but we could not figure out what was meant by "DV". If the reviewer meant that we should indicate the separate peaks for the grip force and the load force in the figure, we disagree, because we are afraid that it will reduce the clarity of the figure.

Results

Please include the p value for the non-significant 10ms peak grip force at 10ms (and other such non-significant results throughout)

We added for the grip force at 10 ms: ($t_{11} = -0.74$, $p = 0.47$).

We added for the others [other materials]: ($t_{11} < 2.2$, $p > 0.052$).

We corrected 3.6 to 3.7

Discussion

The word 'participant' is misspelled (about line 30, left column).

Corrected

The effect outlined in this paragraph does seem to suggest a rather sophisticated mechanism at play though – maybe the authors should discuss this in the context of (Mawase, & Karniel, 2010, Evidence for predictive control in lifting series of virtual objects Experimental brain research 203 (2), 447-452

In our opinion, the experimental conditions in this paper are too different from our experiments, so we decided not to cite this paper.

The time course of this effect is similar to that reported in a recent paper using haptic cues to guide action which maybe deserves some discussion in this context (Pruszynski, J.A., Johansson, R.S., Flanagan, J.R. (2016). A rapid tactile-motor reflex automatically guides reaching towards handheld objects. Current Biology 26: 788-792.)

Thank you for directing us to this recent paper. We added the following text to the Discussion:

A similar result was found in a recent study in another area of sensorimotor control, the planning and execution of motion. In an experiment where participants had to reach towards a target that could either stay still or move sideways during the motion, no difference in correction latency was found between a condition in which only haptic information was present during the move versus a condition in which both visual and haptic information were present [25]. Similar to the present study, this shows that visual information does not play a large role when haptic information is available.

We added the following reference:

[25] Pruszynski JA, Johansson RS, Flanagan JR. A Rapid Tactile-Motor Reflex Automatically Guides Reaching toward Handheld Objects. Current Biology. 2016;26:788–792.

A further alternate explanation for the interaction with visual condition, which the authors perhaps don't fully acknowledge, is simply order effects – participants have had far more practice by the time they interact with visual cues.

We agree. We added the following part of a sentence to the Discussion:

or because they had had more practice by the time they reached the sighted condition.

Reviewer: 5

Comments to the Author(s)

The manuscript by Bergmann Tiest and Kappers is well written and the topic is interesting and not well studied yet.

Thank you.

However, I have a few comments that, when addressed, hopefully improve the manuscript:

1. Most importantly, I agree with reviewer 1 that the friction measurement by a single author is not sufficient. Friction is not the property of an object it depends on TWO surfaces, here the finger and the material. Thus, the friction will also be determined by the surface properties of the hand (e.g. the moisture) and as the authors state themselves the relation between the two is not linear. One might assume that for some materials, such as sandpaper, the friction coefficient will always be high (because the material is designed in that way), but without measurements we don't know this for sure and especially not for the other materials. Furthermore, the current measurements show large variability between repetitions (as also pointed out by reviewer 1) and they were conducted by the experimenter, not a naïve person. Because there is only data from one person, we have no idea about the interindividual differences or how friction might change over the course of an experiment.

In my opinion, the authors should either measure friction for each participant individually (and then use the rank order of friction coefficients for their stats or simply look at correlations) or restrain from the analyses related to friction. However, this would cut a substantial and interesting part of the manuscript. The remaining results, except analysis of load force (and comparison to grip force) were already presented in the author's 2014 conference paper. Individually determined friction coefficients would clearly make the results and conclusions of the paper much stronger.

At this stage it is impossible to add friction measurements for individual participants to our manuscript. Reviewer 3 also made a remark about the friction measurements (comment 2), but fortunately, s/he found our argument why we did it our way acceptable. We added the following text to the Discussion:

In the present study, CoFs were not measured for each participant separately, but only with the experimenter's finger, so as to be able to pool the data per material. Another way to do this would have been to measure the friction per participant separately, but average the data over participants. This was not done, as averaging across participants would cause a decrease in signal-to-noise ratio.

2. *The authors state that they cannot determine load force prior to movement onset, because they use position data of the infrared marker to do so. Additionally they could use data from the force sensor under the platform to determine load force until lift-off. This may improve the manuscript.*

Unfortunately, we are not able to follow this suggestion, as the force sensor under the platform was not calibrated.

3. *Determination of lift-off: This should be described more clearly. Was it only based on the platform sensor? If yes, what was the criterion? Or was this determined for each signal separately? Were the different measurements (force sensors, position data) not in sync and triggered at the same time?*

We extended the text as follows:

For each trial, the two force profiles were synchronised by identifying the moment of lift-off in the grip force and load force profiles separately. In the grip force profile derived from the force sensors, ...

Later in the same paragraph, we added the word "load":

In the load force profile calculated ...

4. *The authors should give more details about the practice trials, how many were they? Vision or no vision condition etc.*

We extended the description of the practice trials in 3.1.4:

Before the experiment proper began, the blindfolded participant was allowed one or two practice trials using the same stimuli as used in the experiment. Having been exposed to some of the stimuli before the experiment probably reduced the sensorimotor effect associated with the material-weight-illusion during the experiment.

5. *What is the difference between metallic-like and wood-like plastic film? They seem to be quite similar. Do the films have any properties similar to wood and metal? The way the materials are described and named throughout the manuscript and in the figures suggests that they have something in common with real wood and real metal. It should be made clear that those are two different types of plastic (or three counting the acrylic surface).*

We added the following text to subsection 2.1:

The metallic-like and wood-like textures were plastic self-adhesive films mimicking the structure of metal and finished wood, respectively.

6. *The supplementary material should be labeled and explained more clearly. In its current form, it is not possible to understand the files and their content. A proper description of which file is which and what the different columns and rows stand for etc. would be helpful.*

We fully agree with your comment. At the time we submitted our manuscript, we had no time to document these files properly. We now added a descriptive file called `file_list.txt`, documenting every single file in the supplementary material.

Appendix C

Dear editors,

Thank you for once again a round of reviews.

Below follows a summary of the history of this manuscript:

- 11-12-2017 Submission
- 20-02-2018 First reviews. Reviewer 1 stated "an excellently-written manuscript" and "Probably the best-written manuscript I've reviewed in several years". Both Reviewer 1 and Reviewer 2 had a list of comments.
- 09-03-2018 Submission of Revision 1. We extensively explained how we dealt with the comments of both Reviewer 1 and Reviewer 2 and improved the manuscript accordingly.
- 27-06-2018 Reception of reviews by three NEW reviewers.
- 16-09-2018 Submission of Revision 2. Again we seriously dealt with all the comments of the three reviewers.
- 21-11-2018 Reception of reviews. Reviewer 3 is satisfied with the revision. Reviewer 4 has only one small remaining issue that can be dealt with at the proof stage. Also Reviewer 5 states that we dealt with most of his/her concerns. This reviewer had only one recurring issue that unfortunately, we are not able to solve.

The Subject Editor suggested that to deal with this remaining issue of Reviewer 5 is simply to do friction measurements on some additional participants. However, as we stated in our manuscript (in response to this issue) is "*It should be noted that the CoF for a finger touching a specific material can differ greatly from person to person, and also depends strongly on humidity and other circumstances*" (Section 2.1). Although this is an important issue that seems to argue for individual friction measurements, this sentence is followed by an explanation why we made the choice to measure this for only one participant. We had to make the choice between (A) friction measurements for all participants, but not being able to pool the data per material, and (B) friction measurements for a single person, but being able to pool the data per material. For the intended purpose of our study, option (B) was the preferred choice.

In our opinion, adding friction measurements to this manuscript has no additional value as the circumstances and participants would be quite different. We are not willing to add measurements that in our eyes would weaken our manuscript as these measurements would not have any explanatory value.

Below we explain how we dealt with the other small issues.

We sincerely hope that since two of the three recent reviewers (Reviewers 3 and 4) were already satisfied with the previous version of the manuscript and since we already dealt with the comments of Reviewers 1 and 2 (who apparently did not assess the revised manuscript) at an earlier stage, you will find our revised manuscript suitable for publication.

With kind regards,

Wouter M. Bergmann Tiest
Astrid M.L. Kappers

Reply to comments

Subject Editor

reviewer 5 has several comments, but it may be simplest to deal with the issue of friction by following the recommendation to "measure the friction coefficients for more (naïve) individuals, so that we are able to assess the interindividual variability for these materials". Collecting more data beyond that is possible but not necessary.

Please read our response in the letter above and our responses to the comments of the reviewers below. We provide arguments why we think that doing friction measurements with new participants is not useful for the current purpose. We dealt with the other small remaining issues.

Reviewer: 4

Comments to the Author(s)

The authors have done a fine job of addressing my comments. My one issue, which can be addressed at the proofs stage, is the use of the phrase 'over-dimensioned' in the clarification added to 3.1.2 - this isn't a term which really makes any sense to me. 'Larger-than-necessary' perhaps?

We followed this suggestion of the reviewer.

Reviewer: 5

Comments to the Author(s)

The authors addressed most of my concerns. However, my main concern is remaining: The fact that friction measurements were only conducted by one of the authors and not every participant. I disagree with the arguments of the authors:

1) It is not impossible to add friction measurements of individual participants. To me it sounds as if the authors took the data from their 2014 conference proceeding and added the friction measurements by one of the authors post-hoc to do an additional (and in principle very interesting) analysis. While I don't generally disagree with that approach and some of the other new analyses are well done, I am still convinced that friction should be measured individually. Again, friction is a property of two surfaces. Thus, if one wants to test the effect of friction, one has to measure friction. If it is not possible to do this for the same participants, that may just mean collecting data of new participants.

Here we partly repeat our response to the editors.

As we already stated in our revised manuscript "the CoF for a finger touching a specific material can differ greatly from person to person, and also depends strongly on humidity and other circumstances". Although this is an important issue that seems to argue for individual friction measurements, this sentence is followed by an explanation why we made the choice to measure this for only one participant. We had to make the choice between (A) friction measurements for all participants, but not being able to pool the data per material, and (B) friction measurements for a single person, but being able to pool the data per material. For the intended purpose of our study, option (B) was the preferred choice.

If we had chosen option (A), we could not have pooled the data per material. However, in that case, we could have analysed the individual friction measurements that relate to the presented data, but that analysis had not our priority nor our preference. In our opinion,

doing such measurements with a new group of participants, has no explanatory value for our current manuscript, so we did not add such measurements. We do not have the contact details of the original participants, so asking them for friction measurements is no option.

2) I also don't agree with their statistical argument. First, currently the author assume that there are no large differences between friction coefficients of different participants (otherwise they would have to measure it individually), why are they then so worried that they could not pool data of different participants across different materials? Even if friction coefficients are not exactly the same, they could be binned or rank-ordered for that specific analysis. After all we would at least know whether (and how much) they differ between persons. Conversely, I think individual measurements would strengthen their argument enormously and some of their analyses are well possible with individual data (e.g. linear fit in figure 9) and would show the effect of friction on grip force very convincingly. I think at the very least the authors should measure the friction coefficients for more (naïve) individuals, so that we are able to assess the interindividual variability for these materials. Additionally the should present data from the literature to support their underlying assumption that the interindividual differences are smaller than the inter-material differences.

We do not make any assumptions about the inter-individual differences (and whether or not these are larger than the inter-material differences) as we have not measured those. In fact, we stated in section 2.1 that the CoFs can differ greatly from person to person. Moreover, as mentioned in our response to comment 1), we think that measuring friction coefficients for naïve participants has little additional value for the current manuscript. Measuring the friction coefficients of the persons who participated in our experiment could be beneficial, but unfortunately is not an option, as we do not have the current contact details of these persons after all these years.

Other comments:

1) How can the lift-off be determined by using a threshold for the sensor underneath the platform, if that sensor was not calibrated?

This is a follow-up question to our response to comment 2 of this reviewer to the previous version of our manuscript.

The sensor was not calibrated to measure force in Newton, but it could detect the difference between whether or not a weight was placed on it.

We added the following sentence to the text in section 3.1.2: "This last sensor was not calibrated, as it was only used as a switch to detect the presence of a stimulus on the platform."

Reviewer: 3

Comments to the Author(s)

Thank you for addressing my points.

We are glad that you are satisfied with our revision.